# Fundamental Limits of Prompt Compression: A Rate-Distortion Framework for Black-Box Language Models

**Alliot Nagle**[*]
UT Austin

**Adway Girish**[*]
EPFL

**Marco Bondaschi**
EPFL

**Michael Gastpar**
EPFL

**Ashok Vardhan Makkuva**[†]
EPFL

**Hyeji Kim**[†]
UT Austin

## Abstract

We formalize the problem of prompt compression for large language models (LLMs) and present a framework to unify token-level prompt compression methods which create *hard prompts* for black-box models. We derive the distortion-rate function for this setup as a linear program, and provide an efficient algorithm to compute this fundamental limit via the dual of the linear program. Using the distortion-rate function as the baseline, we study the performance of existing compression schemes on a synthetic dataset consisting of prompts generated from a Markov chain, natural language queries, and their respective answers. Our empirical analysis demonstrates the criticality of *query-aware* prompt compression, where the compressor has knowledge of the downstream task/query for the black-box LLM. We show that there is a large gap between the performance of current prompt compression methods and the optimal strategy, and propose *Adaptive QuerySelect*, a query-aware, variable-rate adaptation of a prior work to close the gap. We extend our experiments to a small natural language dataset to further confirm our findings on our synthetic dataset.

## 1 Introduction

In spite of the recent success of transformer-based [1] large language models (LLMs) in language modeling tasks, inference calls to a transformer can be costly in both time and memory usage. Although significant progress has been made to improve the memory usage and runtime efficiency via implementation-level optimizations [2, 3, 4] and architecture-level optimizations and alternatives [5, 6, 7], a third type of optimization that compresses the input (an *input-level* optimization) has the benefit that it directly reduces the resource usage of an LLM inference call, *and* it can be used in conjunction with the other two types of optimizations for further efficiency gains. *In this work, we offer a framework and analysis for a recent body of literature in this direction, known as prompt compression* [8, 9, 10].

The goal of a prompt compression method is to transform a sequence of input tokens $x$ into a shorter sequence of tokens $m$ such that the response generated by a target LLM will semantically mean the same thing regardless of whether $x$ or $m$ is given as input. Using $m$ as the input directly decreases the memory and runtime requirements necessary for an LLM inference call. Moreover, the additional benefits to this approach are: (1) redundant or superfluous tokens are removed, making room to fit more pertinent information in the target LLM's limited-size context window, (2) it can be used in addition to implementation and architecture-level optimizations to get further efficiency gains, and (3) it is the only technique available when seeking to lower costs for black-box API calls to closed-source

---

[*]Equal contribution. [†]Equal contribution.

38th Conference on Neural Information Processing Systems (NeurIPS 2024).

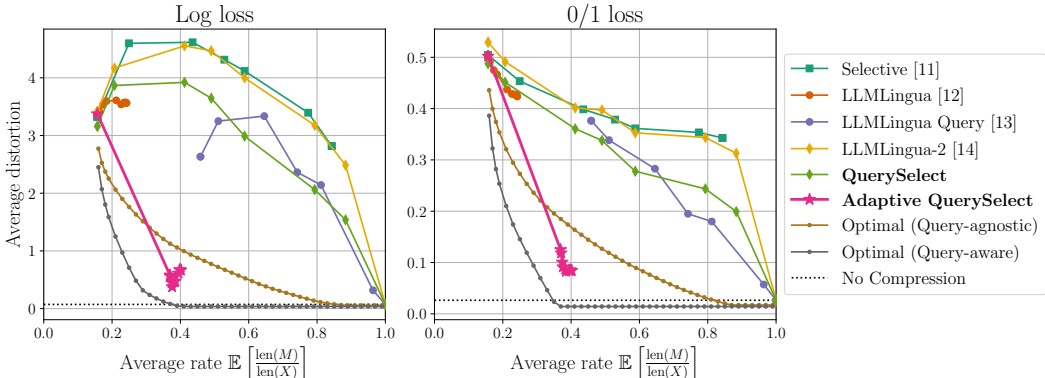

Figure 1: The distortion-rate trade-off of all prompt compression methods compared to the query-aware and query-agnostic theoretical limits on a synthetic dataset with binary prompts. All distortions are computed with the log loss **(left)** and 0/1 loss **(right)** distortion metrics formally defined in (1). We observe that (1) most existing methods are far from the theoretical limit, suggesting that there is still room for improvement in this field, (2) conditioning on the query allows for a significant improvement, as seen by the performance of the query-aware method QuerySelect against the query-agnostic LLMLingua-2 [14], and (3) our proposed method Adaptive QuerySelect, a query-aware and variable-rate adaptation of LLMLingua-2, achieves the best performance among all methods considered, and is the only method to outperform the optimal query-agnostic strategy.

models. This third point is particularly important, since the associated cost for a black-box API model inference call, from the perspective of the caller, is determined by the runtime and the number of input tokens, both which can be reduced with prompt compression. In our framework and analysis, we focus on the *prompt compression for black-box models* setting, where the output of a prompt compression method is a set of tokens ("hard prompts") [11, 12, 13, 14], and exclude methods which output embedding vectors ("soft prompts") [15, 16, 17] as those are not transferable to black-box models.

Despite the progress in the prompt compression literature, there is a lack of proper formalization of this problem and there is no clear framework to unify these works. Most works propose methods that work well but offer no insight into key questions, such as *"How far are we from the theoretical limit of the rate-distortion trade-off?"*, *"How essential is the conditioning on the query when compressing the prompt?"*, and *"How does tokenization impact the performance of prompt compression methods?"* We offer a unifying framework for the problem of prompt compression and seek to answer these questions with theory and experiments. Our main contributions can be summarized as follows.

1. ***Theoretical analysis:*** We formalize the problem of prompt compression and formulate it as a rate-distortion problem (Sec. 3.1). We characterize the optimal trade-off between the rate of compression and the distortion incurred, i.e., the *distortion-rate function*, via a dual linear program, and provide a geometric algorithm to compute this optimal trade-off (Sec. 3.2, Sec. 3.3).

2. ***Evaluation:*** We introduce a synthetic dataset with binary prompts and natural language queries, for which we can compute the distortion-rate function (Sec. 4.1), and compare and obtain insights on existing prompt compression algorithms as in Fig. 1 (Sec. 4.2). We further confirm our findings by extending our experiments to a small natural language dataset and NarrativeQA [18].

3. ***Algorithm design:*** Our novel method, "Adaptive QuerySelect," a query-aware, variable-rate adaptation of LLMLingua-2 [14], outperforms all prompt compression methods on our datasets and has a rate-distortion curve that significantly reduces the gap with the theoretical limit (Sec. 4).

## 2 Background and related works

Long prompts slow the inference process due to the increase in the number of tokens for the LLM to process. It is also known that very long prompts can cause LLMs to "forget" parts of the input and produce erroneous answers [19]. Therefore, studying how these prompts can be compressed is essential. As shown in Fig. 2, we wish to design a compressor that, upon receiving the prompt, produces a "compressed" version (which has fewer tokens than the prompt) called the *compressed*

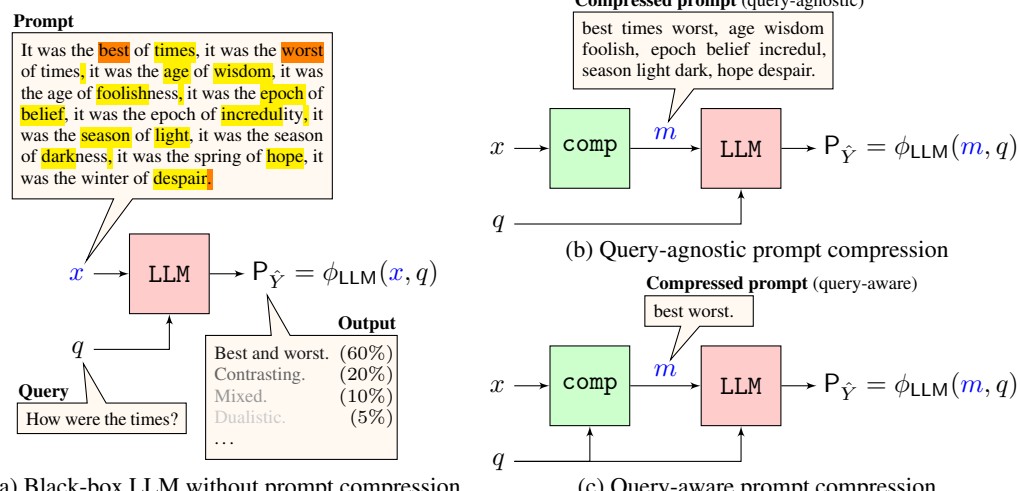

Figure 2: Model for prompt compression in LLMs. (a): Without prompt compression, the LLM takes a long **Prompt** and **Query** as input, and produces an **Output** distribution. (b) and (c): The prompt is passed through a compressor to obtain a shorter **Compressed prompt** and the LLM takes this compressed prompt and query as input instead. (b) The compressor does not have access to the query, and preserves all highlighted tokens. (c) The compressor has access to the query, and preserves only the tokens highlighted in orange.

*prompt*, such that a target LLM is able to give answers that are "close enough," per some appropriately chosen metric, to the ground truth. Though similar in spirit to text summarization, prompt compression has the advantage that the compressed prompt is not required to be human-readable.

All prompt compression methods belong to one of two groups: those that compress the prompt into *soft prompts* and those that compress the prompt into *hard prompts*. In soft-prompt compression, the compressor is trained to transform the input prompt into a set of embedding vectors (sometimes referred to as "soft tokens") that do not map back into the token space. These methods, including Gist Tokens [15], AutoCompressor [17], and In-Context Auto-Encoder [16] are trained end-to-end and require specialized fine-tuning of the target LLM to interpret the soft prompt inputs.

In this work, we focus instead on methods that compress the prompt into hard prompts, where the compressor's output is a set of tokens. While it is technically feasible to fine-tune the target LLM in this setting, it is unnecessary and often avoided because the utility of this setting is compressing prompts for black-box models that are not fine-tuned. These methods often use either the target LLM, or a smaller and faster LLM, to compress the prompt. The basic idea behind all these methods is to identify the tokens that are "most relevant," per an appropriate metric, and retain as many of them in the compressed prompt as possible. These methods include Selective Context [11], LLMLingua [12], LLMLingua-2 [14], and LongLLMLingua [13]. More details on these works can be found in Sec. 4.2. Precursors to the prompt compression works include text compression methods, which have the added constraint that the compressed text is human-readable [20, 21, 22]. Prompt compression methods are different from these in that the text only needs to be interpretable by the target LLM, not by a human.

We offer a framework for hard-prompt compression methods where we assume that a query is provided in addition to the compressed prompt during the target LLM inference call. Functionally, this is the most useful interpretation of prompt compression since it clarifies that the goal is to compress the prompt for a given query/task. This setting is also used in the LLMLingua and LongLLMLingua works, and is more general than the setting where no query is used (the query can then be empty).

## 3   Distortion-rate function for prompt compression

We first formalize the problem of prompt compression, and then develop a rate-distortion framework to study its fundamental limits. In particular, we define and characterize the *distortion-rate function*, which describes the optimal trade-off between how much and how well the prompt is compressed. A complete overview of the notation can be found in App. A.

### 3.1 A formal model for prompt compression

**Black-box LLM.** As depicted in Fig. 2a, we assume that we have a pretrained LLM which takes a pair of the *prompt* $x \in \mathcal{V}^{n_x}$ and the *query* $q \in \mathcal{V}^{n_q}$, $(x, q) \in \mathcal{V}^{n_x + n_q}$ as inputs, where $\mathcal{V}$ refers to the vocabulary of the LLM (i.e., the set of all tokens), and $n_x$ and $n_q$ are the lengths of the prompt and query respectively. The *output* of the LLM is given by $\mathsf{P}_{\hat{Y}} = \phi_{\mathsf{LLM}}(x, q)$, where $\phi_{\mathsf{LLM}} : \mathcal{V}^* \to \mathscr{P}(\mathcal{V}^*)$ is a deterministic function which maps a sequence of tokens to a probability distribution on sequences of tokens. We denote the set of all prompts $x$ by $\mathcal{X}$ and the set of all queries $q$ by $\mathcal{Q}$. Clearly, they are both equal to $\mathcal{V}^*$, but this notation is useful in the subsequent discussion. We model prompt-query pairs $(X, Q)$ as random variables drawn according to the joint distribution $\mathsf{P}_{XQ} \in \mathscr{P}(\mathcal{X} \times \mathcal{Q})$.

In cases where we have a correct answer $y \in \mathcal{Y} = \mathcal{V}^*$ corresponding to the pair $(x, q)$, we characterize the "closeness" of the LLM output $\mathsf{P}_{\hat{Y}} = \phi_{\mathsf{LLM}}(x, q)$ to the answer $y$ using a distortion measure $\mathsf{d} : \mathcal{Y} \times \mathscr{P}(\mathcal{Y}) \to [0, \infty]$. Two possible choices of $\mathsf{d}$ are the log loss $\mathsf{d}_{\log}$ and the 0/1 loss $\mathsf{d}_{0/1}$, given by

$$\mathsf{d}_{\log}(y, \mathsf{P}_{\hat{Y}}) = \log \frac{1}{\mathsf{P}_{\hat{Y}}(y)} \quad \text{and} \quad \mathsf{d}_{0/1}(y, \mathsf{P}_{\hat{Y}}) = \mathbb{1}\left\{ y \neq \underset{\hat{y}}{\arg\max}\, \mathsf{P}_{\hat{Y}}(\hat{y}) \right\}. \tag{1}$$

These are respectively the cross-entropy loss between the distributions $\delta_y$ and $\mathsf{P}_{\hat{Y}}$, and the prediction error. When dealing with natural language queries, a semantic distortion metric such as RougeL [23] or BertScore [24] is more appropriate. Additionally, there is no single answer that is uniquely correct. To account for this variability in what qualifies as a correct answer, we model the *answer* as a random variable $Y$ drawn from the distribution $\mathsf{P}_{Y|XQ}(\cdot|x, q)$, which depends on the prompt $x$ and query $q$. This induces the joint distribution $\mathsf{P}_{XQY} = \mathsf{P}_{XQ}\mathsf{P}_{Y|XQ}$. We characterize the "closeness" between the correct answer and the LLM output by the average distortion, given by $\mathbb{E}_{Y \sim \mathsf{P}_{Y|XQ}(\cdot|x,q)}\left[\mathsf{d}\big(Y, \phi_{\mathsf{LLM}}(x, q)\big)\right]$. With $\mathsf{d} = \mathsf{d}_{\log}$, this is the cross-entropy loss between $\mathsf{P}_{Y|XQ}(\cdot|x, q)$ and $\mathsf{P}_{\hat{Y}} = \phi_{\mathsf{LLM}}(x, q)$, and with $\mathsf{d} = \mathsf{d}_{0/1}$, this is the prediction error probability.

**Prompt compression.** As described in Sec. 2, we consider two types of prompt compression: query-agnostic and query-aware. Fig. 2b depicts the *query-agnostic* version, where the goal is to design a *compressor* denoted by comp as a possibly random function from $\mathcal{X}$ to $\mathcal{M}$, i.e., the set of all compressed prompts. The compressor takes in the prompt $X \sim \mathsf{P}_X$ and produces a *compressed prompt* $M = \mathsf{comp}(X)$ with $\mathrm{len}(M) \leq \mathrm{len}(X)$. Then, the user replaces $X$ with the compressed prompt $M$ and provides the LLM with the input $(M, Q)$, resulting in the output distribution $\mathsf{P}_{\hat{Y}} = \phi_{\mathsf{LLM}}(M, Q)$. To quantify the performance of this compressor comp, two quantities are of interest:

(1) the *rate* $\mathbb{E}\left[\frac{\mathrm{len}(M)}{\mathrm{len}(X)}\right]$, to measure *how much* the prompt is compressed, and

(2) the *distortion* $\mathbb{E}\left[\mathsf{d}\big(Y, \phi_{\mathsf{LLM}}(M, Q)\big)\right]$, to measure *how well* the prompt is compressed,

with both expectations taken with respect to (w.r.t.) $\mathsf{P}_{MXQY}$. If we compress $x$ to a low rate, the compressed prompt $m$ may not retain the information in $x$ that is necessary for the query $q$, leading to an output $\phi_{\mathsf{LLM}}(m, q)$ that is different from $\mathsf{P}_{Y|XQ}(\cdot|x, q)$ and hence, a high distortion. Thus, there is a trade-off between these quantities, which we formalize as the distortion-rate function in Sec. 3.2.

We can also model *query-aware* prompt compression similarly, with the difference being that the compressor also has access to the query $q \in \mathcal{Q}$, as shown in Fig. 2c. In addition to the average rate and distortion computed over all queries, it is also interesting to consider the rate and distortion *for each query*. To simplify the presentation, we restrict our discussion here to the query-agnostic setting, and only briefly mention the analogous definitions and results for the query-aware setting. A complete development of the query-aware setting can be found in App. B.

### 3.2 Rate-distortion formulation for prompt compression

**Distortion-rate function $D^*(R)$.** The *distortion-rate function* for any compression problem characterizes the fundamental trade-off between the distortion and the rate [25, 24, 26, 27]. We say that the pair $(R, D)$ is *achievable* if there exists a compressor with rate at most $R$ and distortion at most $D$. For a given rate $R$, the distortion-rate function $D^*(R)$ is the smallest distortion that can be achieved by a compressor with rate at most $R$. Formally, it is defined as

$$D^*(R) \triangleq \inf\{D \geq 0 \mid (R, D) \text{ is achievable}\}$$
$$= \inf\{D \geq 0 \mid \text{there exists a compressor with rate} \leq R \text{ and distortion} \leq D\}. \tag{2}$$

We are now ready to characterize the distortion-rate function for prompt compression.

$D^*(R)$ **for query-agnostic prompt compression.** Recall that our quantities of interest are the rate $\mathbb{E}\left[\frac{\text{len}(M)}{\text{len}(X)}\right]$, and the distortion $\mathbb{E}\left[\text{d}\big(Y, \phi_{\text{LLM}}(M,Q)\big)\right]$, with both expectations taken w.r.t. $\mathsf{P}_{MXQY}$, where $M = \text{comp}(X)$ for a random function comp. By the functional representation lemma [27, 28], a random function from $\mathcal{X}$ to $\mathcal{M}$ is equivalent to a conditional distribution $\mathsf{P}_{M|X}$. Thus, we can equivalently model the compressor as a conditional distribution $\mathsf{P}_{M|X}$, and (2) is explicitly written as

$$
\begin{aligned}
D^*(R) = \quad &\inf_{\mathsf{P}_{M|X}} \quad \mathbb{E}\left[\text{d}\big(Y, \phi_{\text{LLM}}(M,Q)\big)\right] \\
&\text{s.t.} \quad \mathsf{P}_{M|X} \text{ is a compressor, and} \\
&\qquad\quad \mathbb{E}\left[\frac{\text{len}(M)}{\text{len}(X)}\right] \leq R,
\end{aligned}
\tag{3}
$$

with both expectations taken w.r.t. the joint distribution $\mathsf{P}_{MXQY} = \mathsf{P}_{M|X}\mathsf{P}_{XQY}$ induced by the compressor $\mathsf{P}_{M|X}$. The constraint "$\mathsf{P}_{M|X}$ is a compressor" is short for the following requirements: (1) it is a conditional distribution, i.e., for each $x \in \mathcal{X}$, $\sum_{m \in \mathcal{M}} \mathsf{P}_{M|X}(m|x) = 1$, (2) if $\text{len}(m) > \text{len}(x)$, then $\mathsf{P}_{M|X}(m|x) = 0$, and (3) if $\text{len}(m) = \text{len}(x)$, then $\mathsf{P}_{M|X}(m|x) = 0$ unless $m = x$. This means that the compressor either strictly reduces the length of the prompt or does no compression and retains the original prompt.

Note that all of the expressions in the objective and the constraints in (3) are linear in $\mathsf{P}_{M|X}$. Hence, the optimization problem is simply a linear program (LP), which is simple from an optimization perspective [29, 30]. However, the dimensions of this problem are still large and solving the LP directly quickly becomes infeasible as the lengths of the prompts increase. In Sec. 3.3, we deal with this optimization problem directly, and show that the dual of the LP provides an exact, practically realizable solution.

The extension to the query-aware setting is straightforward; we then have query-dependent (or *conditional*) distortion-rate functions $D_q^*(R)$ for each $q \in \mathcal{Q}$, and an *average* distortion-rate function, denoted by $\bar{D}^*(R)$. Refer to App. B for an explicit characterization of $D_q^*(R)$ and $\bar{D}^*(R)$.

**Connections to information-theoretic setups.** We provide a brief overview of rate-distortion theory from the information theory literature in App. D and describe how our model compares. In particular, we note that our model for prompt compression closely resembles the setup of compression with side-information for function computation [31, 32, 33, 34], where both the encoder and the decoder are part of the system design. More recently, there has also been a growing interest in computing the distortion-rate functions of these classical setups for real-world datasets [35, 36, 37]. However, in our model for prompt compression, only the encoder (which is the compressor) can be designed, hence our model is one of compression *for a fixed decoder*. Such a model has not been actively studied in the information theory literature before, but in the next subsection, we show that the distortion-rate function can be written as an explicit LP in terms of this fixed decoder.

## 3.3 Linear program formulation of the distortion-rate function

Having expressed the distortion-rate function for prompt compression as an LP, we now look to solve this LP. We first rewrite (3) as an explicit LP using optimization-theoretic notation, and hide the probabilistic notation involving expectations and conditional probabilities in the parameters of the LP. Refer to App. A for an overview of the notation.

**Proposition 1** (Primal LP). *The distortion-rate function for query-agnostic prompt compression* (3) *is given by the solution to the linear program*

$$
\begin{aligned}
D^*(R) = \inf_{\left(\boldsymbol{z}_x \in \mathbb{R}_+^{\mathcal{M}_x}\right)_{x \in \mathcal{X}}} \quad &\sum_{x \in \mathcal{X}} \boldsymbol{D}_x^\top \boldsymbol{z}_x \\
\text{s.t.} \quad &\sum_{x \in \mathcal{X}} \boldsymbol{R}_x^\top \boldsymbol{z}_x \leq R, \quad \mathbf{1}^\top \boldsymbol{z}_x = 1, \quad \forall x \in \mathcal{X},
\end{aligned}
\tag{LP}
$$

*where for each $x \in \mathcal{X}$, $\mathcal{M}_x$ denotes the set of compressed prompts associated to $x$, i.e., the set of all possible token sequences of length smaller than $\text{len}(x)$, the vectors $\boldsymbol{z}_x \in \mathbb{R}_+^{\mathcal{M}_x}$ are the optimization*

*variables and the constants $\boldsymbol{D}_x, \boldsymbol{R}_x \in \mathbb{R}_+^{\mathcal{M}_x}$ with components indexed by $m \in \mathcal{M}_x$ are given by*

$$\boldsymbol{D}_{x,m} \triangleq \mathsf{P}_X(x)\, \mathbb{E}\left[\mathsf{d}(Y, \phi_{\mathsf{LLM}}(m, Q))\right] \quad \text{and} \quad \boldsymbol{R}_{x,m} \triangleq \mathsf{P}_X(x)\, \frac{\mathsf{len}(m)}{\mathsf{len}(x)}, \qquad m \in \mathcal{M}_x, \quad (4)$$

*with the expectation taken with respect to $\mathsf{P}_{QY|MX}(\cdot, \cdot | m, x)$.*

*Proof.* This follows immediately from (3) by defining the constants $\boldsymbol{D}_x, \boldsymbol{R}_x \in \mathbb{R}_+^{\mathcal{M}_x}$ for each $x \in \mathcal{X}$ as given in (4), and taking $\boldsymbol{z}_x$ to be $\mathsf{P}_{M|X}(\cdot | x)$. We use the fact that $\mathsf{P}_{M|X}(m | x) = 0$ when $\mathsf{len}(m) > \mathsf{len}(x)$ to reduce the dimension of $\boldsymbol{z}_x$ from $\mathcal{M}$ to $\mathcal{M}_x$ to obtain (LP). ◻

For our experimental setup in Sec. 4.2, we see that dimension of the LP is too large to solve (LP) directly using off-the-shelf solvers. Fortunately, the dual of the LP can be written more concisely, and can also be solved using a relatively simple algorithm.

**Theorem 1** (Dual LP). *The distortion-rate function for query-agnostic prompt compression (3) is given by the solution to the dual of the linear program (LP), i.e.,*

$$D^*(R) = \sup_{\lambda \geq 0} \left\{ -\lambda R + \sum_{x \in \mathcal{X}} \min_{m \in \mathcal{M}_x} \left[ \boldsymbol{D}_{x,m} + \lambda \boldsymbol{R}_{x,m} \right] \right\}. \qquad \text{(dual-LP)}$$

*Proof sketch.* This follows by taking the dual [29] of the LP (LP) and simplifying the resulting expression. For a complete proof, refer to App. C.1. ◻

**Algorithm to solve (dual-LP).** While the optimization problem in (dual-LP) seems difficult to solve, with its max-min structure and the supremum over a continuous variable, it provides a neat geometric interpretation which allows for a computationally simple algorithm given in Algorithm 1. It takes as input $R$, $(\boldsymbol{D}_x)_{x \in \mathcal{X}}$, $(\boldsymbol{R}_x)_{x \in \mathcal{X}}$, and returns as output the distortion-rate function at $R$, i.e., $D^*(R)$. In App. C.2, we prove that the output is indeed $D^*(R)$, and provide a step-by-step illustration of the algorithm on an artificial example. A high-level description of the algorithm is given below.

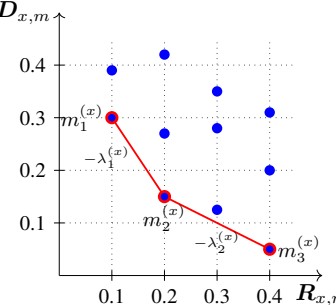

Figure 3: Lower-left convex envelope for an example with $|\mathcal{M}_x| = 11$, $k_x = 3$.

Before presenting the algorithm, it is useful to define the following geometric object. The *lower-left convex envelope* of a set of points in $\mathbb{R}_+^2$ is the largest convex function that lies below and to the left of the points, as shown in Fig. 3 for the points $\{(\boldsymbol{R}_{x,m}, \boldsymbol{D}_{x,m})\}_{m \in \mathcal{M}_x}$ for a fixed $x \in \mathcal{X}$. Let $k_x$ be the number of points on this envelope, then these $k_x$ points are exactly the minimizers of $\min_{m \in \mathcal{M}_x} \left[ \boldsymbol{D}_{x,m} + \lambda \boldsymbol{R}_{x,m} \right]$ for some $\lambda \geq 0$. Solving this inner minimization problem of (dual-LP) is thus easy, and amounts to simply finding the points labelled as "$m^{(x)}$" on the lower-left convex envelope, ordered from left to right, as done in Lines 3–4 of Algorithm 1. Let the magnitudes of slopes of the line segments on the envelope be given by the "$\lambda^{(x)}$" terms in decreasing order (Lines 5–6). Also letting $\lambda_0^{(x)} = +\infty$ and $\lambda_{k_x}^{(x)} = 0$, observe that for $i = 1, \ldots, k_x$, $m_i^{(x)}$ minimizes $\boldsymbol{D}_{x,m} + \lambda \boldsymbol{R}_{x,m}$ over $m$ for $\lambda \in \left[ \lambda_i^{(x)}, \lambda_{i-1}^{(x)} \right)$. Importantly, it is enough to consider just these sequences "$m^{(x)}$" and "$\lambda^{(x)}$" sequences computed for all $x \in \mathcal{X}$ (Lines 2–6) to solve (dual-LP), instead of the entire set $\mathcal{M}_x$ and the continuum of all positive real numbers $\lambda$ respectively. This makes the problem tractable even for large values of $|\mathcal{M}_x|$.

The rest of the algorithm computes the outer supremum. Lines 7–9 prepare for this by introducing new notation "$\widetilde{m}^{(x)}$" and "$\widetilde{\lambda}$" such that for $\lambda \in \left[ \widetilde{\lambda}_j, \widetilde{\lambda}_{j-1} \right)$, $\widetilde{m}_j^{(x)}$ minimizes $\boldsymbol{D}_{x,m} + \lambda \boldsymbol{R}_{x,m}$ over $m \in \mathcal{M}_x$. There is no calculation involved in this step; what we gain is that the range of $\lambda$ on which the minimizer is $\widetilde{m}_j^{(x)}$ no longer depends on $x$. This gives us everything we need to compute the distortion-rate function, which is obtained by lines 10–13. Observe that the input $R$ is only used for lines 10–13, so for a given dataset, lines 1–9 can be run once and the results "$\widetilde{m}^{(x)}$" and "$\widetilde{\lambda}$" stored, with only lines 10–13 run for each value of $R$.

We derive a similar dual LP formulation of the query-aware distortion-rate functions in App. B. In fact, we see that both the conditional and average distortion-rate functions are of the same form as (dual-LP), with different parameters. Hence, Algorithm 1 can compute all of the distortion-rate functions that we have defined, namely the query-agnostic and query-aware (conditional and average) distortion-rate functions.

**Algorithm 1:** To compute the distortion-rate function via the dual linear program (dual-LP)

1 **Input:** $R, (\boldsymbol{D}_x)_{x \in \mathcal{X}}, (\boldsymbol{R}_x)_{x \in \mathcal{X}}$;             **Output:** $D^*(R)$, the distortion-rate function at rate $R$;

2 **for** $x \in \mathcal{X}$ **do**

3      Find $\mathcal{M}_{\text{env}}^{(x)} \subseteq \mathcal{M}_x$ such that $\{(\boldsymbol{R}_{x,m}, \boldsymbol{D}_{x,m})\}_{m \in \mathcal{M}_{\text{env}}^{(x)}}$ are on the lower-left convex boundary of
     $\{(\boldsymbol{R}_{x,m}, \boldsymbol{D}_{x,m})\}_{m \in \mathcal{M}_x}$ ;             ▷ see Fig. 3 for an example

4      $\left\{ m_1^{(x)}, m_2^{(x)}, \ldots, m_{k_x}^{(x)} \right\} \leftarrow \mathcal{M}_{\text{env}}^{(x)}$ ordered such that $\boldsymbol{R}_{x, m_{k_x}^{(x)}} > \cdots > \boldsymbol{R}_{x, m_1^{(x)}}$;

5      **for** $i = 1, \ldots, k_x - 1$ **do** $\lambda_i^{(x)} \leftarrow \frac{\boldsymbol{D}_{x, m_i^{(x)}} - \boldsymbol{D}_{x, m_{i+1}^{(x)}}}{\boldsymbol{R}_{x, m_{i+1}^{(x)}} - \boldsymbol{R}_{x, m_i^{(x)}}}$;

6      $\lambda_0^{(x)} \leftarrow +\infty$; $\lambda_{k_x}^{(x)} \leftarrow 0$; $\Lambda^{(x)} \leftarrow \left\{ \lambda_0^{(x)}, \lambda_1^{(x)}, \ldots, \lambda_{k_x-1}^{(x)}, \lambda_{k_x}^{(x)} \right\}$ ;             ▷ $\lambda_0^{(x)} > \cdots > \lambda_{k_x}^{(x)}$

7 $\left\{ \widetilde{\lambda}_0, \ldots, \widetilde{\lambda}_k \right\} \leftarrow \bigcup_{x \in \mathcal{X}} \Lambda^{(x)}$ with $+\infty = \widetilde{\lambda}_0 > \widetilde{\lambda}_1 > \cdots > \widetilde{\lambda}_{k-1} > \widetilde{\lambda}_k = 0$ ;     ▷ $k \geq k_x \; \forall \, x \in \mathcal{X}$

8 **for** $x \in \mathcal{X}$ **do**

9      **for** $j = 1, \ldots, k$ **do** Find $i \in \{1, \ldots, k_x\} : \left( \lambda_i^{(x)}, \lambda_{i-1}^{(x)} \right) \supseteq \left( \widetilde{\lambda}_j, \widetilde{\lambda}_{j-1} \right)$; set $\widetilde{m}_j^{(x)} \leftarrow m_i^{(x)}$ ;

10 **for** $j = 1, \ldots, k$ **do**

11      **if** $\sum_{x \in \mathcal{X}} \boldsymbol{R}_{x, \widetilde{m}_j^{(x)}} > R$ **then** $\lambda_j \leftarrow \widetilde{\lambda}_{j-1}$ **else** $\lambda_j \leftarrow \widetilde{\lambda}_j$ ;

12      $D_j \leftarrow -\lambda_j R + \sum_{x \in \mathcal{X}} \left[ \boldsymbol{D}_{x, \widetilde{m}_j^{(x)}} + \lambda_j \boldsymbol{R}_{x, \widetilde{m}_j^{(x)}} \right]$;

13 Return $\max_{j=1, \ldots, k} D_j$ ;             ▷ $= D^*(R)$

## 4 Experiments

The distortion-rate function defined in Sec. 3 describes the best possible trade-off between the achievable values of rate and distortion in the query-aware and query-agnostic cases. In this section, we compare the performance of existing prompt compression methods (that are compatible with the black-box model setting we consider here) with the optimal curve for a synthetic dataset. We observe that there is *a sizeable gap* between the performance of existing methods and the optimal curve. We propose Adaptive QuerySelect, a *query-aware and variable-rate adaptation* of LLMLingua-2 [14], that outperforms the existing methods on this synthetic dataset. We also consider a query-aware version of LLMLingua-2 called QuerySelect and observe that it outperforms the query-agnostic version, which highlights the *importance of conditioning on the query*.

We include an ablation study on the impact of tokenization of the prompt compression problem, as tokenization is lossy since it groups together multiple symbols into a single symbol before passing it to an LLM. We study the effect of tokenization on the prompt compression problem by forcing the tokenizer on the encoder and decoder side to tokenize the bits of the binary string prompts in our dataset individually, which we refer to as "forced tokenization." We run experiments in this setting and with the regular "standard tokenization." Additional details on our experiments can be found in App. F. Our code is made available for reproducibility purposes.[2]

### 4.1 Experimental setup

**Dataset.** In order to run experiments that are computationally tractable but still meaningful to the prompt compression problem, we construct a synthetic dataset $\{(x_i, q_i, y_i)\}_{i=1}^N$ with (1) prompts $x_i$ being sequences from $\mathcal{V} = \{0, 1\}$, i.e., binary prompts, (2) natural language queries $q_i$, such as "Count the number of 1s," "Compute the parity," and "Is the binary string a palindrome?" and (3) their associated answers $y_i$. In total, we construct a dataset of seven queries; a complete specification of the dataset, including a few examples is available in App. F.2.1. The binary prompts are generated from a first-order Markov chain on $\{0, 1\}$ with a 0.1 probability of transitioning and a 0.9 probability of remaining in the same state, and the minimum and maximum possible lengths for each prompt are four and ten, respectively. All methods are evaluated on a validation set of 1400 examples in total (7 queries, 200 examples per query). The optimal distortion-rate function is computed using Algorithm 1,

---

[2]Our code is available at `https://github.com/acnagle/fundamental-limits`

taking $\mathsf{P}_{XQY}$ to be the empirical distribution on the dataset, i.e., $\mathsf{P}_{XQY} = \frac{1}{N}\sum_{i=1}^{N}\delta_{(x_i,q_i,y_i)}$. This is the natural choice when the true distribution is unknown. Another choice is a parametric model with parameters learned from a dataset, but it is unclear what is an appropriate model in this case.

We also run experiments on a small natural language dataset curated with GPT-4 [38] and NarrativeQA [18] for a large-scale experiment. The small dataset consists of ten prompts with four queries each, and a few examples are provided in Table 3 in App. F.2.1. More details on the considerations made in constructing our datasets are provided in App. E.

**Baseline methods.** We compare the rate-distortion trade-off of the optimal strategy (both query-aware and query-agnostic) with prompt compression methods that can be used to compress prompts for a black-box target LLM: Selective Context [11], LLMLingua [12], LLMLingua Query [13], LLMLingua-2 [14]. As such, we do not consider methods like Gist Tokens [15], In-Context Autoencoder [16], and AutoCompressor [17] since they require special training methods generally not compatible with black-box target LLMs. Selective Context uses $-\log \mathsf{P}(x_i \mid x_0, x_1, \ldots, x_{i-1})$ to score the $i$-th token, and retains the tokens whose score is larger than the $p$-percentile, where $p \in [0, 1]$ is the ratio parameter. LLMLingua uses a similar method, but they first partition the input prompt into segments and condition on previously compressed segments to compress the current segment. They later extended their method to perform query-aware compression, which is what we use for LLMLingua Query. While these methods use a decoder-style (causal) transformer LLM to do prompt compression, this approach makes an independence assumption on the influence of future tokens have on the $i$-th token. LLMLingua-2 instead uses an encoder-style (bidirectional) LLM to perform a token classification task, where their model predicts whether a given token should be kept or removed.

**Our proposed methods.** We add two novel contributions over the LLMLingua-2 work: (1) we adapt LLMLingua-2 to the query-aware setting, whereas the original work only proposed the query-agnostic approach, which we call "QuerySelect," and (2) we further adapt this query-aware approach into a *variable-rate* approach we refer to as "Adaptive QuerySelect." This approach lets the encoder model decide which tokens to keep based on the confidence over a specified threshold. In other words, LLMLingua-2 and QuerySelect accept a rate parameter to determine the compression ratio, but Adaptive QuerySelect replaces the rate parameter with a threshold parameter. The encoder model predicts the probability of keeping a particular token, and the token is kept if the predicted probability is above the threshold, resulting in a variable-rate compression of the prompt. Variable-rate compression is important as some prompts are more compressible than others, and vice versa.

**Models.** We use Mistral 7B Instruct v0.2 [39] as our black-box target LLM, which is fine-tuned on the training set partition of our synthetic dataset. This model is fixed after fine-tuning and no prompt compression methods have access to any part of it. All prompt compression methods use an LLM as part of their compression algorithm; we use deduplicated Pythia 1B [40] for Selective Context, LLMLingua, and LLMLingua Query and RoBERTa Base [41] for LLMLingua-2-based methods. For each method, we finetune on the training set partition to enable the best performance possible for that method. More information on how we trained these methods and the data we used is in App. F. For all models, including the target LLM, we fine-tune with LoRA [42] and conduct a hyperparameter grid search. We choose the configuration with the best performance on a test set that is different from the validation set. More details on the hyperparameter search are provided in App. F.3.

For the natural language dataset, no fine-tuning is necessary on the decoder side. On the encoder side, Selective Context, LLMLingua, and LLMLingua Query use the same model as on the decoder side, and LLMLingua-2 uses a specially fine-tuned version of XLM RoBERTa Large [43, 14]. We use a custom fine-tuned XLM RoBERTa Large model as the encoder for the QuerySelect and Adaptive QuerySelect methods. The training dataset of (prompt, query, answer) tuples used to train this custom model is filtered from the Databricks Dolly 15k [44] dataset to only include examples with prompt lengths between a specified minimum and maximum length (see Sec. F.3.2 for details).

## 4.2 Results

Fig. 1 summarizes our experimental contributions on the synthetic dataset. We observe a *large gap* between the optimal curve and existing prompt compression methods. Thus, we propose QuerySelect as a *query-aware* and Adaptive QuerySelect as a *query-aware, variable-rate modification* of LLMLingua-2 to close this gap. Our results show that Adaptive QuerySelect achieves the best performance and, in fact, is the only method to outperform the optimal query-agnostic strategy. We

also note that the optimal distortion-rate curves eventually fall below the baseline performance of using the full prompt (no compression). This observation is especially interesting because it shows that compressing prompts can improve performance on downstream tasks, as observed on natural language datasets in previous prompt compression works [12, 13, 14]. We accredit the performance of Adaptive QuerySelect to variable-rate compression, where we allow the compressor to choose how much it should compress based on the query and prompt as input (see App. B, Remark 1 for a formal explanation of variable-rate compression). Even though this approach relinquishes explicit control over the rate, our experiments show that variable-rate compression is the closest to optimality.

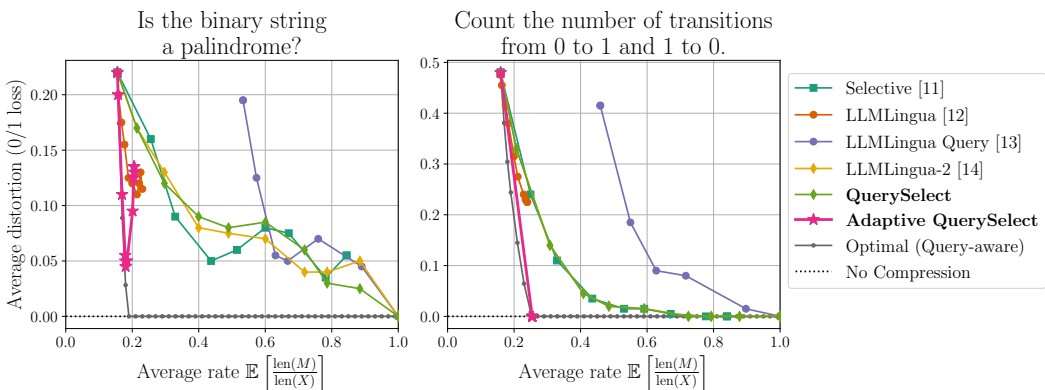

Figure 4: We highlight the distortion-rate curves for two of the seven queries in the validation partition of our synthetic dataset. Our method, Adaptive QuerySelect, is able to match the performance of the optimal query-aware strategy **(left)**. Some queries naturally incur less distortion than others with the target LLM, even with a query-agnostic approach, if the query is aligned well with the data generation process for the prompt **(right)**. Note that QuerySelect covers the line of LLMLingua-2 as their performance is identical for this query.

**Gap from optimality depends on the query.** In Fig. 4, we highlight the distortion-rate curves for two out of seven of the queries in our synthetic dataset. Despite the fact that Fig. 1 shows a gap in average performance between the query-aware optimal strategy and Adaptive QuerySelect, Fig. 4 **(left)** shows that Adaptive QuerySelect can match the performance of the optimal query-aware compression scheme. Comparing Fig. 4 **(left)** and **(right)**, we see that the prompt compression problem is easier (methods are closer to optimality) for certain tasks or queries depending on how the prompts were generated. For our synthetic dataset, all prompts are generated from a Markov chain with a transition probability of 0.1 and a probability of 0.9 for remaining in the same state. This means the tokens with the highest entropy are those that are part of a transition, and those tokens are the most important for answering this query. As a result, we see that methods that use the negative log-likelihood as a means for compression (Selective Context, LLMLingua, and LLMLingua Query) perform well, even without conditioning on the query. An exception here is the performance of LLMLingua Query, which we find has mixed performance compared to vanilla LLMLingua for token-level prompt compression on our dataset. Please refer to Fig. 11 in App. F for results on all queries.

**Effect of tokenization.** Finally, the results of our ablation study on the effects of tokenization are provided in Fig. 10 in App. F. Interestingly, the optimal curves are nearly identical, suggesting that tokenization does not play a role in attaining the best possible trade-off. Furthermore, we see that, for a fixed rate, the standard tokenization performance often matches or exceeds the performance of forced tokenization. However, the standard tokenization approach does not allow for average rates below 0.6 due to the limited size of the prompts in our synthetic dataset, so the comparison is somewhat limited. In particular, standard tokenization allows for compression of at most four tokens (but usually only two or three tokens), whereas forced tokenization allows for compression of at most ten tokens.

**Extension to natural language prompts.** We have thus far shown the gap between current token-level prompt compression algorithms and their optimal strategies for both the query-aware and query-agnostic encoder on a synthetic dataset. Here, we extend our results to a small natural language dataset curated with GPT-4 [38] (details in App. F.2.1). For natural language prompts, the number of possible combinations of tokens grows too quickly, either by increasing the number of tokens in the prompt or increasing the vocabulary size, to compute the full $\boldsymbol{D}_x$. Instead, we rely on the observation

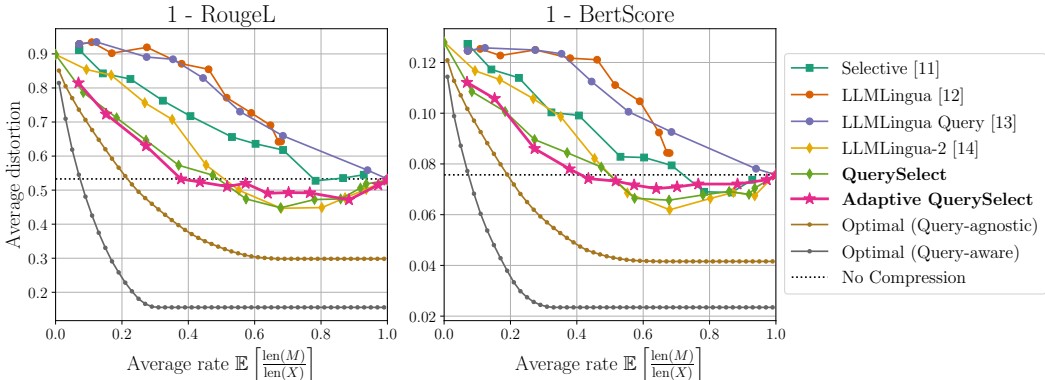

Figure 5: Comparison among all prompt compression methods on our natural language dataset. We show the rate-distortion trade-off for RougeL [23] **(left)** and BertScore [45] **(right)**. Since a higher RougeL and BertScore metric is better, we plot "1− the computed average distortion" so that a higher rate should yield a lower loss. We discuss the choice of our metrics in App. F.2.2.

that current token-level prompt compression strategies simply remove tokens in place. With this observation, the number of prompts to consider has the same growth rate as a prompt with a binary alphabet. Please refer to App. E for more details. Although we cannot compute the true optimal curves where every possible combination of tokens is considered, we *can* compute the optimal curves for current prompt compression algorithms that do not generate new tokens. Fig. 8 shows that the gap for this approximation is negligible on binary prompts.

The results of our extension to natural language prompts, presented in Fig. 5, show that both of our proposed methods achieve the lowest distortion among all other prompt compression algorithms for low rates. However, the gap between all algorithms and the optimal strategies is significant. We posit the quality of the training data for LLMLingua-2-based methods accounts for the discrepancy in how far away the best method is from the optimal strategies between binary (Fig. 1) and natural language (Fig. 5) prompts. More specifically, the labels used for the binary synthetic dataset can be determined algorithmically and are optimal, but GPT-4 is used to determine the labels for the natural language dataset, which generally does not have a set of optimal "ground truth" labels. Remarkably, the gap between feeding the prompt directly to the black-box LLM (no compression) and either optimal prompt compression strategy is also large, and Fig. 5 shows that much lower distortion can be achieved in roughly 70% and 40% fewer tokens for the query-aware and query-agnostic cases, respectively. LLMLingua-2 methods are the only methods that achieve lower distortion than the no compression result, albeit for higher rates. Finally, we present a few histograms of the rates for QuerySelect and Adaptive QuerySelect in Fig. 15, which shows the greater range of rates that Adaptive QuerySelect may choose from over QuerySelect.

Although we cannot compute the optimal rate-distortion curves on a dataset as large as NarrativeQA, we did compute the curve for all existing methods to compare them on a larger-scale dataset. Those results are provided in Fig. 16; they confirm that our proposed methods outperform all other methods for rates below 0.5. We also display the average time to compress a prompt for each method in Table 6. Since our methods are adapted from LLMLingua-2, they share the same runtime.

## 5 Conclusion

We have proposed a framework for understanding the prompt compression problem for black-box target LLMs. With this framework, we defined and formulated the optimal distortion-rate trade-off as a linear program, and devised an algorithm to solve this efficiently via its dual, for both query-agnostic and query-aware settings. We compared the optimal curves with prompt compression methods in the existing literature and adapt one of them, LLMLingua-2, to be query-aware and variable-rate; this modified method, Adaptive QuerySelect, exhibits superior performance, sometimes even matching the performance of the optimal query-aware strategy, on our synthetic dataset. As future work, it is important to exhaustively study our proposed method on natural language datasets. Additionally, it is worthwhile to pursue an approximation to the optimal curves for large-scale datasets to observe the fundamental limit in that regime. We share preliminary results in that direction in App. G.3.

## Acknowledgments

This work was partly supported by ARO Award W911NF2310062, ONR Award N000142412542, and the 6G@UT center within WNCG at UT Austin. The work was also supported in part by the Swiss National Science Foundation under Grant 200364. The authors would like to thank Ananda Theertha Suresh for introducing them to the problem of prompt compression. AG would like to thank Emre Telatar for helpful discussions on the problem formulation.

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

# Appendix

The appendix is organized as follows:

## A Notation

**General.** We use $\triangleq$ to signify a definition. For a set $\mathcal{X}$, with $x_i \in \mathcal{X}$ for $i = 1, \ldots, n$, we represent the sequence $(x_1, \ldots, x_n)$ by $x^n \in \mathcal{X}^n$, which is short for $\mathcal{X} \times \cdots \times \mathcal{X}$. We use $\mathcal{X}^*$ to denote $\bigcup_{n \geq 1} \mathcal{X}^n$, the set of all nonempty finite-length sequences on $\mathcal{X}$. In general, for $y \in \mathcal{X}^n$, we denote the length of $y$ by $\text{len}(y) = n$. We denote the cardinality of a set $\mathcal{X}$ by $|\mathcal{X}|$. We use $\mathbb{R}_+$ to denote the set of nonnegative real numbers. For a set $\mathcal{X} = \{x_1, \ldots, x_k\}$, we use the boldface $(\boldsymbol{A}_x)_{x \in \mathcal{X}}$ to denote the vector $(\boldsymbol{A}_{x_1}, \ldots, \boldsymbol{A}_{x_k})$ indexed by elements of $\mathcal{X}$. We also write $\mathbb{R}_+^{\mathcal{X}} = \{(\boldsymbol{v}_x)_{x \in \mathcal{X}} : \boldsymbol{v}_x \in \mathbb{R}_+ \text{ for each } x \in \mathcal{X}\}$. We use $\mathbb{1}$ to denote the indicator function, which takes the value 1 when its argument is true and 0 otherwise. The infimum and supremum of a set of values is denoted using $\inf$ and $\sup$ respectively. We use $\boldsymbol{0}$ and $\boldsymbol{1}$ to denote the vector of appropriate dimension with all elements equal to 0 and 1 respectively.

**Probability.** We deal with discrete probability distributions on finite sets, for which we use calligraphic letters to denote the set (e.g. $\mathcal{X}$), uppercase letters to denote the random variable (r.v., e.g. $X$) and lowercase letters to denote samples of the r.v. (e.g. $x$). The set of all probability distributions on the set $\mathcal{X}$ is denoted by $\mathscr{P}(\mathcal{X})$. The probability distribution of the r.v. $X$ on $\mathcal{X}$ is denoted by $\mathsf{P}_X \in \mathscr{P}(\mathcal{X})$, and we say $X \sim \mathsf{P}_X$. For a (measurable) function $f$, the expectation of the r.v. $f(X)$ is denoted by $\mathbb{E}_{X \sim \mathsf{P}_X}[f(X)]$, or $\mathbb{E}[f(X)]$, with the subscripts dropped when the distribution and/or r.v.'s are clear from context. The degenerate probability distribution with mass 1 at $x \in \mathcal{X}$ is represented by $\delta_x \in \mathscr{P}(\mathcal{X})$. Conditional probabilities from $\mathcal{X}$ to $\mathcal{Y}$ are denoted as $\mathsf{P}_{Y|X}$, and for each $x \in \mathcal{X}$, we denote the distribution on $Y$ as $\mathsf{P}_{Y|X}(\cdot|x)$.

**Problem-setup-specific notation.** In our model, we use $\mathcal{V}$ to refer to the vocabulary of the prompt. We use the uppercase letters $X$ to refer to the prompt, $M$ to refer to the compressed prompt, $Q$ to refer to the query and $Y$ to refer to the answer as random variables (and corresponding calligraphic and lowercase letters to denote the set and samples of the r.v. respectively). We use $\mathsf{P}_{\hat{Y}}$ to refer to the output distribution of the LLM, which is modeled as the function $\phi_{\mathsf{LLM}}$. For a given prompt $x$, we use $\mathcal{M}_x$ to refer to the set of possible compressed prompts, which is the set of all sequences of length smaller than $\text{len}(x)$. To denote the distortion measure, we use $\mathsf{d}$, which can be either the log loss $\mathsf{d}_{\log}$ or the 0/1 loss $\mathsf{d}_{0/1}$. We denote the query-agnostic distortion-rate function at rate $R$ by $D^*(R)$. The average query-aware distortion-rate function is denoted by $\bar{D}^*(R)$, and the conditional query-aware distortion-rate function for query $q \in \mathcal{Q}$ is given by $D_q^*(R)$.

## B Extensions to query-aware prompt compression

As mentioned in Sec. 3.2 and Sec. 3.3, analogous definitions and results can be obtained for the query-aware setting as well. The difference is that the compressor has access to the query in addition

to the prompt. Thus, the compressor comp is a possibly random function from $\mathcal{X} \times \mathcal{Q}$ to $\mathcal{M}$. For the query $Q$, the compressor maps the prompt $X$ to the compressed prompt $M = \texttt{comp}(X, Q)$ with $\text{len}(M) \leq \text{len}(X)$. The user provides the input $[M, Q]$ to the LLM, which produces the output distribution $\mathsf{P}_{\hat{Y}} = \phi_{\mathsf{LLM}}(M, Q)$. Just as in the query-agnostic setting, two quantities of interest are

$$\text{(1) the (average) rate } \mathbb{E}\left[\frac{\text{len}(M)}{\text{len}(X)}\right], \quad \text{and} \quad \text{(2) the distortion } \mathbb{E}\left[\mathsf{d}_{\log}(Y, \phi_{\mathsf{LLM}}(M, Q))\right],$$

with both expectations taken with respect to the joint distribution $\mathsf{P}_{MXQY}$. Since different queries may require different amounts of information to be preserved during compression, it is also of interest to define the (conditional) rate and distortion *for the specific query $q$* as $\mathbb{E}\left[\frac{\text{len}(M)}{\text{len}(X)}\right]$ and $\mathbb{E}\left[\mathsf{d}_{\log}(Y, \phi_{\mathsf{LLM}}(M, q))\right]$ respectively, with both expectations taken with respect to the joint distribution $\mathsf{P}_{MXY|Q}(\cdot|q)$.

The rate-distortion problem for query-aware prompt compression can be also formulated similarly to (3). We model comp as a random mapping $\mathsf{P}_{M|XQ}$ from $\mathcal{X} \times \mathcal{Q}$ to $\mathcal{M}$. Then, the (average) distortion-rate function at rate $R$ is the smallest distortion that can be achieved by a query-aware compressor with rate at most $R$, given by

$$\bar{D}^*(R) = \inf_{\mathsf{P}_{M|XQ}} \mathbb{E}\left[\mathsf{d}_{\log}(Y, \phi_{\mathsf{LLM}}(M, Q))\right]$$
$$\text{s.t.} \quad \mathsf{P}_{M|XQ} \text{ is a compressor, and}$$
$$\mathbb{E}\left[\frac{\text{len}(M)}{\text{len}(X)}\right] \leq R, \tag{5}$$

with both expectations taken with respect to the joint distribution $\mathsf{P}_{MXQY} = \mathsf{P}_{M|XQ}\mathsf{P}_{XQY}$ induced by the compressor. The condition "$\mathsf{P}_{M|XQ}$ is a compressor" is short for (1) for each $x \in \mathcal{X}$ and $q \in \mathcal{Q}$, $\sum_{m \in \mathcal{M}} \mathsf{P}_{M|XQ}(m|x, q) = 1$, (2) $\mathsf{P}_{M|XQ}(m|x, q) = 0$ if $\text{len}(m) > \text{len}(x)$, and (3) if $\text{len}(m) = \text{len}(x)$, then $\mathsf{P}_{M|XQ}(m|x, q) = 0$ unless $m = x$. Similarly, the (conditional) distortion-rate function at rate $R$ is the smallest distortion that can be achieved by a query-aware compressor for query $q$ at rate at most $R$, given by

$$D_q^*(R) = \inf_{\mathsf{P}_{M|XQ}(\cdot|\cdot, q)} \mathbb{E}\left[\mathsf{d}_{\log}(Y, \phi_{\mathsf{LLM}}(M, Q))\right]$$
$$\text{s.t.} \quad \mathsf{P}_{M|XQ}(\cdot|\cdot, q) \text{ is a compressor, and}$$
$$\mathbb{E}\left[\frac{\text{len}(M)}{\text{len}(X)}\right] \leq R, \tag{6}$$

with both expectations taken with respect to $\mathsf{P}_{MXY|Q}(\cdot, \cdot, \cdot|q)$.

Just like the query-agnostic setting, note that both (5) and (6) are linear programs, as the objective and constraints are all linear in $\mathsf{P}_{M|XQ}$ and $\mathsf{P}_{M|XQ}(\cdot|\cdot, q)$ respectively. We obtain explicit linear programs analogous to (LP) by defining constants $\bar{\boldsymbol{D}}_x^q$ and $\bar{\boldsymbol{R}}_x^q \in \mathbb{R}_+^{\mathcal{M}_x}$ for the average distortion-rate function and $\boldsymbol{D}_x^q$ and $\boldsymbol{R}_x^q \in \mathbb{R}_+^{\mathcal{M}_x}$ for the conditional distortion-rate functions, for each $x \in \mathcal{X}$ and $q \in \mathcal{Q}$, similarly to (4).

**Proposition 2** (Query-aware primal LPs). *The (average) distortion-rate function for query-aware prompt compression* (5) *is given by the solution to*

$$\bar{D}^*(R) = \inf_{\left(\boldsymbol{z}_{x,q} \in \mathbb{R}_+^{\mathcal{M}_x}\right)_{x \in \mathcal{X}, q \in \mathcal{Q}}} \sum_{x \in \mathcal{X}, q \in \mathcal{Q}} \bar{\boldsymbol{D}}_x^{q\top} \boldsymbol{z}_{x,q}$$
$$\text{s.t.} \quad \sum_{x \in \mathcal{X}, q \in \mathcal{Q}} \bar{\boldsymbol{R}}_x^{q\top} \boldsymbol{z}_{x,q} \leq R, \quad \text{(avg-cond-LP)}$$
$$\mathbf{1}^\top \boldsymbol{z}_{x,q} = 1, \quad \forall x \in \mathcal{X}, q \in \mathcal{Q}.$$

*The (conditional) distortion-rate function for query-aware prompt compression for query q (6) is given by the solution to*

$$D_q^*(R) = \inf_{(\boldsymbol{z}_x \in \mathbb{R}_+^{\mathcal{M}_x})_{x \in \mathcal{X}}} \quad \sum_{x \in \mathcal{X}} \boldsymbol{D}_x^{q\top} \boldsymbol{z}_x$$

$$s.t. \quad \sum_{x \in \mathcal{X}} \boldsymbol{R}_x^{q\top} \boldsymbol{z}_x \leq R, \qquad \text{(cond-LP)}$$

$$\mathbf{1}^\top \boldsymbol{z}_x = 1, \quad \forall\, x \in \mathcal{X}.$$

*For each $x \in \mathcal{X}$, $\mathcal{M}_x$ denotes the set of all possible compressed prompts associated to $x$, i.e., the set of all possible token sequences of length at most $\mathrm{len}(x)$, the vectors $\boldsymbol{z}_{x,q} \in \mathbb{R}_+^{\mathcal{M}_x}$ for $q \in \mathcal{Q}$ and $\boldsymbol{z}_x \in \mathbb{R}_+^{\mathcal{M}_x}$ are the optimization variables respectively and the constants $\bar{\boldsymbol{D}}_x^q, \bar{\boldsymbol{R}}_x^q, \boldsymbol{D}_x^q, \boldsymbol{R}_x^q \in \mathbb{R}_+^{\mathcal{M}_x}$ are given by*

$$\bar{\boldsymbol{D}}_{x,m}^q \triangleq \mathsf{P}_{XQ}(x,q)\, \mathbb{E}\left[\mathsf{d}_{\log}(Y, \phi_{\mathsf{LLM}}(m,q))\right] \;\; and \;\; \bar{\boldsymbol{R}}_{x,m}^q \triangleq \mathsf{P}_{XQ}(x,q)\frac{\mathrm{len}(m)}{\mathrm{len}(x)}, \quad m \in \mathcal{M}_x, \;\; (7)$$

$$\boldsymbol{D}_{x,m}^q \triangleq \mathsf{P}_{X|Q}(x|q)\, \mathbb{E}\left[\mathsf{d}_{\log}(Y, \phi_{\mathsf{LLM}}(m,q))\right] \;\; and \;\; \boldsymbol{R}_{x,m}^q \triangleq \mathsf{P}_{X|Q}(x|q)\frac{\mathrm{len}(m)}{\mathrm{len}(x)}, \quad m \in \mathcal{M}_x, \;\; (8)$$

*with the expectation taken with respect to $\mathsf{P}_{Y|MXQ}(\cdot|m,x,q)$.*

*Proof.* This follows immediately from (5) and (6) by defining the constants $\bar{\boldsymbol{D}}_x^q, \bar{\boldsymbol{R}}_x^q, \boldsymbol{D}_x^q, \boldsymbol{R}_x^q \in \mathbb{R}_+^{\mathcal{M}_x}$ for each $x \in \mathcal{X}$ and $q \in \mathcal{Q}$ as given in (7) and (8) and taking $\boldsymbol{z}_{x,q}$ and $\boldsymbol{z}_x$ to be $\mathsf{P}_{M|XQ}(\cdot|x,q)$ respectively,. We use the fact that $\mathsf{P}_{M|XQ}(m|x,q) = 0$ when $\mathrm{len}(m) > \mathrm{len}(x)$ to reduce the dimension of $\boldsymbol{z}_{x,q}$ and $\boldsymbol{z}_x$ from $\mathcal{M}$ to $\mathcal{M}_x$. $\qquad\square$

**Remark 1.** An interesting phenomenon here that does not occur in the query-agnostic setting is the comparison between the average and conditional distortion-rate functions, i.e., $\bar{D}^*(R)$ and $D_q^*(R)$ for $q \in \mathcal{Q}$. One possible way to "average" the conditional distortion-rate functions would be to simply compute $\mathbb{E}_{Q \sim \mathsf{P}_Q}\left[D_Q^*(R)\right]$, but we always have $\bar{D}^*(R) \leq \mathbb{E}_{Q \sim \mathsf{P}_Q}\left[D_Q^*(R)\right]$. This is because the latter averages the distortion-rate functions over $\mathsf{P}_Q$ at a fixed value of the rate, i.e., the prompt for each query is forced to be compressed to the same rate $R$. For $\bar{D}^*(R)$, on the other hand, only the *average* rate over the queries is required to be $R$. This allows the compressor to set a higher rate for "difficult queries" that have higher distortion values, and use a lower rate for queries that have lower distortion values in general. This is exactly the phenomenon we exploit in designing the *variable-rate* compression scheme Adaptive QuerySelect in Sec. 4.1, which outperforms other existing schemes in our experiments.

Just as in the query-agnostic setting, it is useful to compute and solve the dual linear programs instead of directly solving the linear programs above.

**Theorem 2** (Query-aware dual LPs). *The (average) distortion-rate function for query-aware prompt compression (5) is given by the solution to the dual of the linear program (avg-cond-LP), i.e.,*

$$\bar{D}^*(R) = \sup_{\lambda \geq 0} \left\{ -\lambda R + \sum_{x \in \mathcal{X}, q \in \mathcal{Q}} \min_{m \in \mathcal{M}_x} \left[ \bar{\boldsymbol{D}}_{x,m}^q + \lambda \bar{\boldsymbol{R}}_{x,m}^q \right] \right\}. \qquad \text{(avg-cond-dual-LP)}$$

*The (conditional) distortion-rate function for query-aware prompt compression for query q (6) is given by the solution to the dual of the linear program (cond-LP), i.e.,*

$$D_q^*(R) = \sup_{\lambda \geq 0} \left\{ -\lambda R + \sum_{x \in \mathcal{X}} \min_{m \in \mathcal{M}_x} \left[ \boldsymbol{D}_{x,m}^q + \lambda \boldsymbol{R}_{x,m}^q \right] \right\}. \qquad \text{(cond-dual-LP)}$$

*Proof.* (Conditional) This follows trivially by simply observing that the linear program in (cond-LP) is identical to that in (LP), except that $\boldsymbol{D}_x$ and $\boldsymbol{R}_x$ are replaced by the (conditional) query-aware versions $\boldsymbol{D}_x^q$ and $\boldsymbol{R}_x^q$ respectively. Henc, by Thm. 1 the solution to (cond-LP) is given by (dual-LP) with $\boldsymbol{D}_x$ and $\boldsymbol{R}_x$ replaced by the (conditional) query-aware versions $\boldsymbol{D}_x^q$ and $\boldsymbol{R}_x^q$ respectively, which gives (cond-dual-LP).

(Average) In addition to replacing $\boldsymbol{D}_x$ and $\boldsymbol{R}_x$ from (LP) by the (average) query-aware versions $\bar{\boldsymbol{D}}_x^q$ and $\bar{\boldsymbol{R}}_x^q$ respectively, we also have that the optimization variables are given by $z_{x,q}$ for each pair $(x, q) \in \mathcal{X} \times \mathcal{Q}$ as opposed to simply $z_x$ for each $x \in \mathcal{X}$. Hence, by Thm. 1 the solution to (avg-cond-LP) is given by (dual-LP) with $\boldsymbol{D}_x$ and $\boldsymbol{R}_x$ are replaced by the (average) query-aware versions $\bar{\boldsymbol{D}}_x^q$ and $\bar{\boldsymbol{R}}_x^q$ respectively and $\mathcal{X}$ replaced by $\mathcal{X} \times \mathcal{Q}$. This gives (avg-cond-dual-LP) exactly, and we are done. □

Note that both (avg-cond-dual-LP) and (cond-dual-LP) are of the same form as (dual-LP), with some minor differences. For a given $q \in \mathcal{Q}$, the conditional distortion-rate function $D_q^*(R)$ is identical to the query-unaware distortion-rate function $D^*(R)$ with $(\boldsymbol{D}_x, \boldsymbol{R}_x)$ replaced by $(\boldsymbol{D}_x^q, \boldsymbol{R}_x^q)$, and hence can be solved by running Algorithm 1 with the input $\left\{R, (\boldsymbol{D}_x^q)_{x \in \mathcal{X}}, (\boldsymbol{R}_x^q)_{x \in \mathcal{X}}\right\}$. For the average distortion-rate function $\bar{D}^*(R)$, in addition to replacing $\boldsymbol{D}_x$ and $\boldsymbol{R}_x$ by $\bar{\boldsymbol{D}}_x^q$ and $\bar{\boldsymbol{R}}_x^q$, we also have that $\mathcal{X}$ is replaced by $\mathcal{X} \times \mathcal{Q}$, hence $\bar{D}^*(R)$ is obtained by running 1 with the input $\left\{R, (\boldsymbol{D}_{x'}^q)_{x' \in \mathcal{X}'}, (\boldsymbol{R}_{x'}^q)_{x' \in \mathcal{X}'}\right\}$, where $\mathcal{X}' \triangleq \mathcal{X} \times \mathcal{Q}$ and $x'$ runs over all pairs $(x, q)$.

## C   The dual linear program: proof and solution

### C.1   Derivation of the dual linear program

*Proof of Thm. 1.* We start from the linear program (LP) and construct its dual. Recall that (LP) is given by

$$D^*(R) = \inf_{\left(\boldsymbol{z}_x \in \mathbb{R}_+^{\mathcal{M}_x}\right)_{x \in \mathcal{X}}} \sum_{x \in \mathcal{X}} \boldsymbol{D}_x^\top \boldsymbol{z}_x$$
$$\text{s.t.} \quad \sum_{x \in \mathcal{X}} \boldsymbol{R}_x^\top \boldsymbol{z}_x \leq R,$$
$$\mathbf{1}^\top \boldsymbol{z}_x = 1, \quad \forall\, x \in \mathcal{X}.$$

Introduce the Lagrange multipliers $\lambda \geq 0$ to handle the inequality constraint and $\mu_x \in \mathbb{R}$ for each $x \in \mathcal{X}$ to handle the equality constraints. Then, the above equation is equivalent to

$$D^*(R) = \inf_{\left(\boldsymbol{z}_x \in \mathbb{R}_+^{\mathcal{M}_x}\right)_{x \in \mathcal{X}}} \left\{ \sum_{x \in \mathcal{X}} \boldsymbol{D}_x^\top \boldsymbol{z}_x + \sup_{\lambda \geq 0} \lambda \left( \sum_{x \in \mathcal{X}} \boldsymbol{R}_x^\top \boldsymbol{z}_x - R \right) + \sum_{x \in \mathcal{X}} \left[ \sup_{\mu_x \in \mathbb{R}} \mu_x \left( \mathbf{1}^\top \boldsymbol{z}_x - 1 \right) \right] \right\}.$$

To see why this equivalence holds, observe that the terms $\sup_{\lambda \geq 0} \lambda \left( \sum_{x \in \mathcal{X}} \boldsymbol{R}_x^\top \boldsymbol{z}_x - R \right)$ and $\sup_{\mu_x \in \mathbb{R}} \mu_x \left( \mathbf{1}^\top \boldsymbol{z}_x - 1 \right)$ are both $0$ when $(\boldsymbol{z}_x)_{x \in \mathcal{X}}$ is in the feasible set of (LP) and $+\infty$ otherwise. Let $\mu \triangleq (\mu_x)_{x \in \mathcal{X}} \in \mathbb{R}^{\mathcal{X}}$, then we can simplify the above expression by rearranging terms, to obtain

$$D^*(R) = \inf_{\left(\boldsymbol{z}_x \in \mathbb{R}_+^{\mathcal{M}_x}\right)_{x \in \mathcal{X}}} \sup_{\substack{\mu \in \mathbb{R}^{\mathcal{X}}, \\ \lambda \geq 0}} \left\{ \sum_{x \in \mathcal{X}} \left( \boldsymbol{D}_x + \lambda \boldsymbol{R}_x + \mu_x \mathbf{1} \right)^\top \boldsymbol{z}_x - \lambda R - \sum_{x \in \mathcal{X}} \mu_x \right\}.$$

Note that the objective $\sum_{x \in \mathcal{X}} \left( \boldsymbol{D}_x + \lambda \boldsymbol{R}_x + \mu_x \mathbf{1} \right)^\top \boldsymbol{z}_x - \lambda R - \sum_{x \in \mathcal{X}} \mu_x$ is linear in $(\boldsymbol{z}_x)_{x \in \mathcal{X}}$ and in $(\mu, \lambda)$, and the minimization and maximization are both over convex sets. Hence, by the minmax theorem [46, 47], we can switch their order without affecting the equality, i.e.,

$$D^*(R) = \sup_{\substack{\mu \in \mathbb{R}^{\mathcal{X}}, \\ \lambda \geq 0}} \inf_{\left(\boldsymbol{z}_x \in \mathbb{R}_+^{\mathcal{M}_x}\right)_{x \in \mathcal{X}}} \left\{ \sum_{x \in \mathcal{X}} \left( \boldsymbol{D}_x + \lambda \boldsymbol{R}_x + \mu_x \mathbf{1} \right)^\top \boldsymbol{z}_x - \lambda R - \sum_{x \in \mathcal{X}} \mu_x \right\}.$$

If, for some $x$, there is a component of the vector $\boldsymbol{D}_x + \lambda \boldsymbol{R}_x + \mu_x \mathbf{1} \in \mathbb{R}^{\mathcal{M}_x}$ that is negative, then letting that component of $\boldsymbol{z}_x$ go to infinity, we have that the inner infimum is $-\infty$. On the other hand, if every component of $\boldsymbol{D}_x + \lambda \boldsymbol{R}_x + \mu_x \mathbf{1}$ is nonnegative for every $x$, then the infimum is simply $0$, attained by setting $\boldsymbol{z}_x = \mathbf{0}$. Hence, the above equation reduces to

$$D^*(R) = \sup_{\substack{\mu \in \mathbb{R}^{\mathcal{X}}, \\ \lambda \geq 0}} -\lambda R - \sum_{x \in \mathcal{X}} \mu_x$$

$$\text{s.t.} \qquad \boldsymbol{D}_{x,m} + \lambda \boldsymbol{R}_{x,m} + \mu_x \geq 0 \quad \text{ for every } m \in \mathcal{M}_x \text{ and } x \in \mathcal{X}.$$

For a given $x$, the constraint $\boldsymbol{D}_{x,m} + \lambda \boldsymbol{R}_{x,m} + \mu_x \geq 0$ for all $m \in \mathcal{M}_x$ is equivalent to $-\mu_x \leq \min_{m \in \mathcal{M}_x} (\boldsymbol{D}_{x,m} + \lambda \boldsymbol{R}_{x,m})$. Letting $\nu_x \triangleq \min_{m \in \mathcal{M}_x} (\boldsymbol{D}_{x,m} + \lambda \boldsymbol{R}_{x,m}) + \mu_x$ and $\nu \triangleq (\nu_x)_{x \in \mathcal{X}}$, the constraint is simply that $\nu_x \geq 0$ for all $x$, or equivalently, $\nu \in \mathbb{R}_+^{\mathcal{X}}$. Hence, the above equation can be written as

$$D^*(R) = \sup_{\substack{\nu \in \mathbb{R}_+^{\mathcal{X}}, \\ \lambda \geq 0}} \; -\lambda R + \sum_{x \in \mathcal{X}} \min_{m \in \mathcal{M}_x} (\boldsymbol{D}_{x,m} + \lambda \boldsymbol{R}_{x,m}) - \sum_{x \in \mathcal{X}} \nu_x.$$

Observe that only the first two terms depend on $\lambda$, and only the last term depends on $\nu$. This lets us optimize over $\lambda$ and $\nu$ separately, to give

$$D^*(R) = \sup_{\lambda \geq 0} \left\{ -\lambda R + \sum_{x \in \mathcal{X}} \min_{m \in \mathcal{M}_x} (\boldsymbol{D}_{x,m} + \lambda \boldsymbol{R}_{x,m}) \right\} + \sup_{\nu \in \mathbb{R}_+^{\mathcal{X}}} \left( -\sum_{x \in \mathcal{X}} \nu_x \right)$$

$$= \sup_{\lambda \geq 0} \left\{ -\lambda R + \sum_{x \in \mathcal{X}} \min_{m \in \mathcal{M}_x} (\boldsymbol{D}_{x,m} + \lambda \boldsymbol{R}_{x,m}) \right\},$$

since $\sup_{\nu \in \mathbb{R}_+^{\mathcal{X}}} \left( -\sum_{x \in \mathcal{X}} \nu_x \right) = -\inf_{\nu \in \mathbb{R}_+^{\mathcal{X}}} \sum_{x \in \mathcal{X}} \nu_x = -\sum_{x \in \mathcal{X}} \inf_{\nu_x \geq 0} \nu_x = 0$, and we are done. $\qquad \square$

### C.2  Proof and illustration of Algorithm 1

In this section, we explain each step of Algorithm 1 in detail. In doing so, we prove that the algorithm does indeed solve (dual-LP), i.e., computes

$$D^*(R) = \sup_{\lambda \geq 0} \left\{ -\lambda R + \sum_{x \in \mathcal{X}} \min_{m \in \mathcal{M}_x} [\boldsymbol{D}_{x,m} + \lambda \boldsymbol{R}_{x,m}] \right\}.$$

We also use an artificial example as described below to show the working of the algorithm, in particular lines 1–9. For convenience, the algorithm is repeated verbatim below, without comments:

---

**Algorithm 1:** To compute the distortion-rate function via the dual linear program (dual-LP)

---

1 **Input:** $R$, $(\boldsymbol{D}_x)_{x \in \mathcal{X}}$, $(\boldsymbol{R}_x)_{x \in \mathcal{X}}$;        **Output:** $D^*(R)$, the distortion-rate function at rate $R$;

2 **for** $x \in \mathcal{X}$ **do**

3      Find $\mathcal{M}_{\mathrm{env}}^{(x)} \subseteq \mathcal{M}_x$ such that $\{(\boldsymbol{R}_{x,m}, \boldsymbol{D}_{x,m})\}_{m \in \mathcal{M}_{\mathrm{env}}^{(x)}}$ are on the lower-left convex boundary of $\{(\boldsymbol{R}_{x,m}, \boldsymbol{D}_{x,m})\}_{m \in \mathcal{M}_x}$;

4      $\left\{ m_1^{(x)}, m_2^{(x)}, \ldots, m_{k_x}^{(x)} \right\} \leftarrow \mathcal{M}_{\mathrm{env}}^{(x)}$ ordered such that $\boldsymbol{R}_{x, m_{k_x}^{(x)}} > \cdots > \boldsymbol{R}_{x, m_1^{(x)}}$;

5      **for** $i = 1, \ldots, k_x - 1$ **do** $\lambda_i^{(x)} \leftarrow \dfrac{\boldsymbol{D}_{x, m_i^{(x)}} - \boldsymbol{D}_{x, m_{i+1}^{(x)}}}{\boldsymbol{R}_{x, m_{i+1}^{(x)}} - \boldsymbol{R}_{x, m_i^{(x)}}}$;

6      $\lambda_0^{(x)} \leftarrow +\infty$; $\lambda_{k_x}^{(x)} \leftarrow 0$; $\Lambda^{(x)} \leftarrow \left\{ \lambda_0^{(x)}, \lambda_1^{(x)}, \ldots, \lambda_{k_x - 1}^{(x)}, \lambda_{k_x}^{(x)} \right\}$;

7 $\left\{ \widetilde{\lambda}_0, \ldots, \widetilde{\lambda}_k \right\} \leftarrow \bigcup_{x \in \mathcal{X}} \Lambda^{(x)}$ with $+\infty = \widetilde{\lambda}_0 > \widetilde{\lambda}_1 > \cdots > \widetilde{\lambda}_{k-1} > \widetilde{\lambda}_k = 0$ ;

8 **for** $x \in \mathcal{X}$ **do**

9      **for** $j = 1, \ldots, k$ **do** Find $i \in \{1, \ldots, k_x\} : \left( \lambda_i^{(x)}, \lambda_{i-1}^{(x)} \right) \supseteq \left( \widetilde{\lambda}_j, \widetilde{\lambda}_{j-1} \right)$;   set $\widetilde{m}_j^{(x)} \leftarrow m_i^{(x)}$ ;

10 **for** $j = 1, \ldots, k$ **do**

11      **if** $\sum_{x \in \mathcal{X}} \boldsymbol{R}_{x, \widetilde{m}_j^{(x)}} > R$ **then** $\lambda_j \leftarrow \widetilde{\lambda}_{j-1}$ **else** $\lambda_j \leftarrow \widetilde{\lambda}_j$ ;

12      $D_j \leftarrow -\lambda_j R + \sum_{x \in \mathcal{X}} \left[ \boldsymbol{D}_{x, \widetilde{m}_j^{(x)}} + \lambda_j \boldsymbol{R}_{x, \widetilde{m}_j^{(x)}} \right]$;

13 Return $\max_{j = 1, \ldots, k} D_j$;

---

Consider the following artificial example with $\mathcal{X} = \{\alpha, \beta\}$. Let $(\mathbf{R}_\alpha, \mathbf{D}_\alpha)$ and $(\mathbf{R}_\beta, \mathbf{D}_\beta)$ be as given by the blue points in the scatter plots over $m \in \mathcal{M}_\alpha$ and $m \in \mathcal{M}_\beta$ respectively in Fig. 6. In our example, we have $|\mathcal{M}_\alpha| = 11$ and $|\mathcal{M}_\beta| = 8$. The following observation is crucial: recall the definitions of $\mathbf{R}_x$ and $\mathbf{D}_x$,

$$\mathbf{D}_{x,m} \triangleq \mathsf{P}_X(x)\, \mathbb{E}\left[\mathsf{d}_{\log}(Y, \phi_{\mathsf{LLM}}(m, Q))\right] \quad \text{and} \quad \mathbf{R}_{x,m} \triangleq \mathsf{P}_X(x)\frac{\mathsf{len}(m)}{\mathsf{len}(x)}, \qquad m \in \mathcal{M}_x,$$

with the expectation computed with respect to $\mathsf{P}_{QY|MX}(\cdot, \cdot | m, x)$. For a fixed value of $x$, the positive real numbers $\mathbf{D}_{x,m}$ can be arbitrary, but $\mathbf{R}_{x,m}$ must be an integral multiple of the constant $\frac{\mathsf{P}_X(x)}{\mathsf{len}(x)}$. Hence, for a given $x$, $\mathbf{R}_{x,m}$ takes at most $\mathsf{len}(x)$ possible values. This turns out to be extremely beneficial in the first step, namely identifying the points on the lower-left convex boundary.

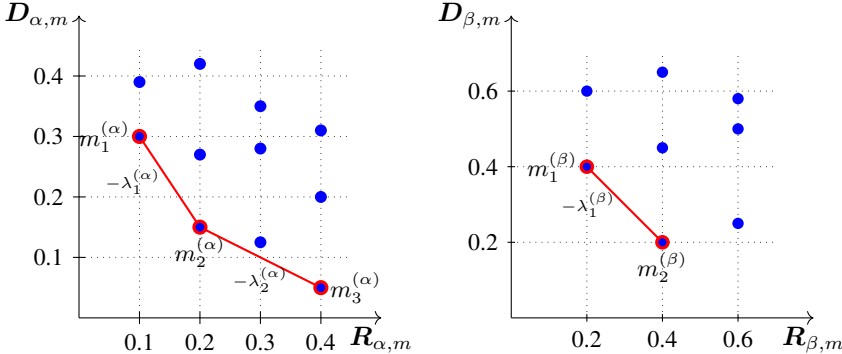

Figure 6: Scatter plots showing the points $\{(\mathbf{R}_{\alpha,m}, \mathbf{D}_{\alpha,m})\}_{m \in \mathcal{M}_\alpha}$ and $\{(\mathbf{R}_{\beta,m}, \mathbf{D}_{\beta,m})\}_{m \in \mathcal{M}_\beta}$ in blue. The associated lower-left convex boundaries $\mathcal{M}_{\mathsf{bd}}^{(\alpha)} = \{m_1^{(\alpha)}, m_2^{(\alpha)}, m_3^{(\alpha)}\}$ and $\mathcal{M}_{\mathsf{bd}}^{(\beta)} = \{m_1^{(\beta)}, m_2^{(\beta)}\}$ are in red; $\lambda_1^{(\alpha)}, \lambda_2^{(\alpha)}$ and $\lambda_1^{(\beta)}$ are the magnitudes of the slopes of the associated line segments.

For $x \in \mathcal{X}$, lines 2–3 of the algorithm identify the $k_x$ points $m_1^{(x)}, \ldots, m_{k_x}^{(x)}$ that lie on the lower-left convex boundary of $\{(\mathbf{R}_{x,m}, \mathbf{D}_{x,m})\}_{m \in \mathcal{M}_x}$. The lower-left convex boundaries are given by the red lines and the points lying on the boundary are outlined in red. Observe that $k_\alpha = 3$ and $k_\beta = 2$. The quantities computed in line 5 are simply the magnitudes of the slopes of the line segments on the boundary. A simple computation gives the result of line 6 of the algorithm in our example to be $\Lambda^{(\alpha)} = \{+\infty, 1.5, 0.5, 0\}$ and $\Lambda^{(\beta)} = \{+\infty, 1, 0\}$. Clearly, for a given value of $x \in \mathcal{X}$ and $\lambda \in \left[\lambda_j^{(x)}, \lambda_{j-1}^{(x)}\right)$, we have that $m_i^{(x)}$ minimizes $\mathbf{D}_{x,m} + \lambda \mathbf{R}_{x,m}$ over all $m \in \mathcal{M}_x$, by virtue of the fact that these points come from the lower-left convex boundary. Hence, for $\lambda \in \left[\lambda_j^{(x)}, \lambda_{j-1}^{(x)}\right)$, we have

$$\sum_{x \in \mathcal{X}} \min_{m \in \mathcal{M}_x} \left[\mathbf{D}_{x,m} + \lambda \mathbf{R}_{x,m}\right] = \sum_{x \in \mathcal{X}} \left[\mathbf{D}_{x,m_j^{(x)}} + \lambda \mathbf{R}_{x,m_j^{(x)}}\right].$$

We cannot simplify this further in its current state since $\lambda$ in the above expression depends on $x$. Hence, we must remove the dependence of the range $\left[\lambda_j^{(x)}, \lambda_{j-1}^{(x)}\right)$ on $x$. To do so, observe that each $\Lambda^{(x)}$ is a partition of $\mathbb{R}_+$ on which $m_i^{(x)}$ is the minimizer of $\mathbf{D}_{x,m} + \lambda \mathbf{R}_{x,m}$. Line 7 of the algorithm simply constructs the union of all these partitions, with $k$ elements, denoted by the $\widetilde{\lambda}$ variables; here we have $k = 4$ and the union is $\{+\infty, 1.5, 1, 0.5, 0\}$. For each $x$, the minimizer on each interval $\left[\widetilde{\lambda}_j, \widetilde{\lambda}_{j-1}\right)$ of the finer partition is known exactly to be one of the $m_i^{(x)}$'s; lines 8–9 associate to each interval the corresponding minimizer, given by $\widetilde{m}_j^{(x)}$. There is no computation involved in these steps, only notational rewriting. The corresponding values obtained for our example are given in the table below (with $\widetilde{\lambda}_0 = +\infty$); observe that $\widetilde{m}_j^{(x)}$ minimizes $\mathbf{D}_{x,m} + \lambda \mathbf{R}_{x,m}$ over $m \in \mathcal{M}_x$ for $\lambda \in \left[\widetilde{\lambda}_j, \widetilde{\lambda}_{j-1}\right)$.

At this point, we have for $\lambda \in \left[\widetilde{\lambda}_j, \widetilde{\lambda}_{j-1}\right)$,

$$\sum_{x \in \mathcal{X}} \min_{m \in \mathcal{M}_x} \left[\mathbf{D}_{x,m} + \lambda \mathbf{R}_{x,m}\right] = \sum_{x \in \mathcal{X}} \left[\mathbf{D}_{x,\widetilde{m}_j^{(x)}} + \lambda \mathbf{R}_{x,\widetilde{m}_j^{(x)}}\right]$$

Table 1: The outputs produced by lines 7–9 of Algorithm 1 with $(\boldsymbol{R}_\alpha, \boldsymbol{D}_\alpha)$ and $(\boldsymbol{R}_\beta, \boldsymbol{D}_\beta)$ as given in Fig. 6.

| $j$ | $\widetilde{\lambda}_j$ | $\widetilde{m}_j^{(\alpha)}$ | $\widetilde{m}_j^{(\beta)}$ |
|---|---|---|---|
| 1 | 1.5 | $m_1^{(\alpha)}$ | $m_1^{(\beta)}$ |
| 2 | 1 | $m_2^{(\alpha)}$ | $m_1^{(\beta)}$ |
| 3 | 0.5 | $m_2^{(\alpha)}$ | $m_2^{(\beta)}$ |
| 4 | 0 | $m_3^{(\alpha)}$ | $m_2^{(\beta)}$ |

$$= \left( \sum_{x \in \mathcal{X}} \boldsymbol{D}_{x, \widetilde{m}_j^{(x)}} \right) + \lambda \left( \sum_{x \in \mathcal{X}} \boldsymbol{R}_{x, \widetilde{m}_j^{(x)}} \right).$$

Hence, the right-hand side of (dual-LP) is simply

$$\max_{j=1,\ldots,k} \sup_{\lambda \in [\widetilde{\lambda}_j, \widetilde{\lambda}_{j-1})} \left\{ \left( \sum_{x \in \mathcal{X}} \boldsymbol{D}_{x, \widetilde{m}_j^{(x)}} \right) + \lambda \left( \sum_{x \in \mathcal{X}} \boldsymbol{R}_{x, \widetilde{m}_j^{(x)}} - R \right) \right\}$$

$$= \max_{j=1,\ldots,k} \left\{ \left( \sum_{x \in \mathcal{X}} \boldsymbol{D}_{x, \widetilde{m}_j^{(x)}} \right) + \sup_{\lambda \in [\widetilde{\lambda}_j, \widetilde{\lambda}_{j-1})} \lambda \left( \sum_{x \in \mathcal{X}} \boldsymbol{R}_{x, \widetilde{m}_j^{(x)}} - R \right) \right\},$$

where the first equality follows since $\{\widetilde{\lambda}_j\}_{j=0}^k$ is a partition of $\mathbb{R}_+$. Consider the term $\sup_{\lambda \in [\widetilde{\lambda}_j, \widetilde{\lambda}_{j-1})} \lambda \left( \sum_{x \in \mathcal{X}} \boldsymbol{R}_{x, \widetilde{m}_j^{(x)}} - R \right)$. If $\boldsymbol{R}_{x, \widetilde{m}_j^{(x)}} > R$, this supremum occurs in the limit as $\lambda \to \widetilde{\lambda}_{j-1}$, otherwise it is achieved at $\lambda = \widetilde{\lambda}_j$. Hence, defining $\lambda_j$ to be $\widetilde{\lambda}_{j-1}$ or $\widetilde{\lambda}_j$ accordingly as in line 11, we have that the above expression is simply $\max_{j=1,\ldots,k} \left\{ \left( \sum_{x \in \mathcal{X}} \boldsymbol{D}_{x, \widetilde{m}_j^{(x)}} \right) + \lambda_j \left( \sum_{x \in \mathcal{X}} \boldsymbol{R}_{x, \widetilde{m}_j^{(x)}} - R \right) \right\}$, which is exactly what line 13 returns. Hence, we have that the algorithm correctly computes the distortion-rate function $D^*(R)$.

## D  Connections to information theory literature

Rate-distortion theory is an area of information theory introduced by Shannon [25] to study the fundamental limits of source compression. The simplest rate-distortion setup is shown in Fig. 7a: We are given a source which generates samples $X_1, \ldots, X_n$ independently and identically distributed (i.i.d.) according to the distribution $\mathsf{P}_X$ on the set $\mathcal{X}$. We are also given a reconstruction alphabet $\hat{\mathcal{X}}$, which may or may not be equal to $\mathcal{X}$. The goal is to compress $X^n$ to a sequence of $k$ elements from an alphabet $\mathcal{V}$, such that a reconstruction onto $\hat{\mathcal{X}}^n$ is as "faithful" as possible, while keeping $k$ as small as possible (in the information theory literature, $\mathcal{V} = \{0, 1\}$ typically). The fidelity of representation is quantified by a *distortion function* $\mathsf{d} : \mathcal{X} \times \hat{\mathcal{X}} \to [0, \infty]$. For example, the problem of compressing images with $p$ real-valued pixels into bit sequences can be cast in this formulation by taking $\mathcal{X} = \hat{\mathcal{X}} = \mathbb{R}^p$, $\mathcal{V} = \{0, 1\}$, and the squared-loss distortion function $\mathsf{d}(x, \hat{x}) = \|x - \hat{x}\|_2^2$.

Formally, the goal is to construct an encoder $\mathrm{enc} : \mathcal{X}^n \to \mathcal{V}^k$ and a decoder $\mathrm{dec} : \mathcal{V}^k \to \hat{\mathcal{X}}^n$ such that: (1) the *rate* $k/n$, and (2) the *(average) distortion* $\mathbb{E}\left[ \mathsf{d}(X^n, \mathrm{dec}(\mathrm{enc}(X^n))) \right]$, are both as small as possible. We say that the rate-distortion pair $(R, D)$ is *achievable* for the source $\mathsf{P}_X$ under the distortion function $\mathsf{d}$ if there exists an $(\mathrm{enc}, \mathrm{dec})$ pair with rate at most $R$ and average distortion at most $D$. If the pair $(R, D)$ is achievable, then clearly, for $\widetilde{R} \geq R$ and $\widetilde{D} \geq D$, the pair $(\widetilde{R}, \widetilde{D})$ is also achievable. Thus, the quantity of interest to us is the lower boundary of the set of achievable $(R, D)$ pairs. This is given by the *distortion-rate function* $D^*$, which is defined as follows: the distortion-rate function at rate $R$ is the smallest distortion $D$ such that the pair $(R, D)$ is achievable, or equivalently,

$$D^*(R) \triangleq \inf\{D \geq 0 \mid (R, D) \text{ is achievable}\} \tag{9}$$
$$= \inf\{D \geq 0 \mid \text{there exists } (\mathrm{enc}, \mathrm{dec}) \text{ with rate} \leq R \text{ and distortion} \leq D\}.$$

Note that the distortion-rate function depends on the source $\mathsf{P}_X$ and the choice of distortion measure. Closed form expressions are known in some cases, the reader is encouraged to refer to classical

texts on information theory [24, 26, 27] for examples. It is important to note that the distortion-rate function is a fundamental limit; no choice of encoder and decoder can give a lower rate and a lower distortion. Thus, the distortion-rate function characterizes the Pareto-optimal front of the trade-off between rate and distortion. It is more common in the information theory literature to define the *rate-distortion function* $R^*(D)$, which is the smallest rate $R$ such that the pair $(R, D)$ is achievable. The two functions trace the same curve when plotted on the same two-dimensional plane.

Several variants of this problem can be defined by introducing the notion of *side-information*, where we have i.i.d. samples $(X_1, Q_1), \ldots, (X_n, Q_n)$ of a pair of correlated random variables $(X, Q) \sim \mathsf{P}_{XQ} \in \mathscr{P}(\mathcal{X} \times \mathcal{Q})$. A natural question to ask is what improvement is possible in terms of the rate-distortion trade-off for $X^n$ when either the encoder or decoder or both have access to this side-information $Q^n$, which is correlated with $X$. If only the encoder has access to $Q^n$, then no improvement can be obtained. If both the encoder and decoder have access to $Q^n$ as shown in Fig. 7c and studied by Gray [48], then, clearly, an improvement is possible. Surprisingly, we can obtain nontrivial improvements when the decoder has access to $Q^n$ as shown in Fig. 7b and studied by Wyner and Ziv [31, 32].

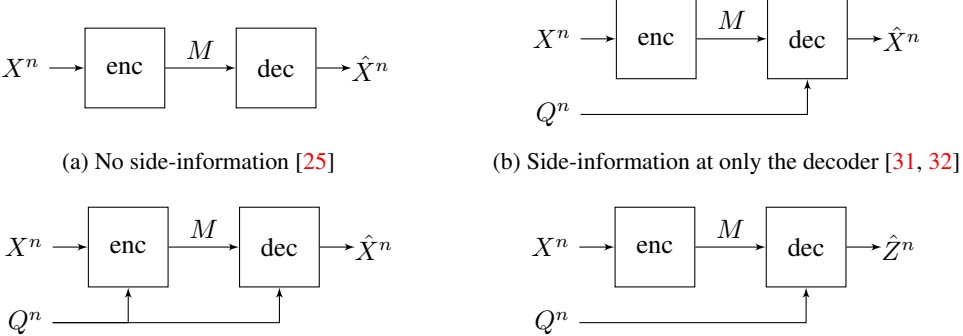

(a) No side-information [25]

(b) Side-information at only the decoder [31, 32]

(c) Side-information at the encoder and the decoder [48]    (d) For function computation, $Z = f(X, Q)$ [34]

Figure 7: Rate-distortion models of compression.

These models resemble our setups for query-aware and query-agnostic prompt compression respectively, with a key difference being that the decoder in our problem is the pretrained LLM, which is fixed. An rate-distortion setup that is closer to our problem in this sense is that of compression for function computation, introduced by [34]. Here, the goal is to recover an estimate $\hat{Z}^n$ that is close to $Z^n$, with $Z_i = f(X_i, Q_i)$ for some desired function $f$. At first glance, it might appear that this setup is exactly our model for prompt compression, but this turns out to be false — the desired output is an estimate of $f(X, Q)$, but the decoder can be designed to compute *any arbitrary function* of $M$ and $Q$. In prompt compression, we have the constraint that the function computed by the decoder is fixed to be $\phi_{\mathsf{LLM}}$, in addition to requiring that the output be close to some function of $X$ and $Q$. Thus, our model for prompt compression actually corresponds to a rate-distortion problem for function computation with side-information *with a fixed decoder*, which has not been studied before, to the best of our knowledge. The distortion-rate function $D^*(R)$ for this setup is given by (LP) and (dual-LP). A closed form expression for $D^*(R)$ cannot be obtained without making further assumptions on $\phi_{\mathsf{LLM}}$; nonetheless, $D^*(R)$ can be computed for any $\phi_{\mathsf{LLM}}$ by solving Algorithm 1.

# E    Extension to natural language datasets

As discussed in Sec. 4.2, we also use Algorithm 1 to compute the distortion-rate function for a small natural language dataset. The decisive bottleneck in running Algorithm 1 turns out to be obtaining $\boldsymbol{D}_x$ for each $x \in \mathcal{X}$, i.e., the input to the algorithm. We require one inference call for each possible compressed prompt $m \in \mathcal{M}_x$ to compute $\boldsymbol{D}_x$ for a particular prompt $x$ and query. Taking $\mathcal{M}_x$ to be all sequences of length smaller than $\mathrm{len}(x)$, we see that that the size of $\mathcal{M}_x$ is $\sum_{i=1}^{\mathrm{len}(x)-1} |\mathcal{V}|^i$, where $\mathcal{V}$ is the vocabulary of the LLM. The model used in our experiments, Mistral 7B Instruct v0.2 [39], has $|\mathcal{V}| = 32,000$. Clearly, it is then virtually impossible to consider prompts with more than 2 tokens, and in fact, makes it difficult to consider even medium length prompts (50 tokens) for a vocabulary of size more than 2.

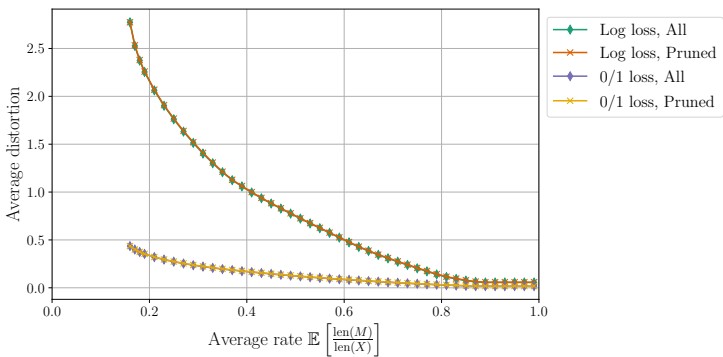

Figure 8: Query-agnostic distortion-rate curves plotted for log loss and 0/1 loss distortion measures. The curves marked with a 'diamond' are computed using all possible shorter sequences, while those marked with an '×' are computed using only pruned versions of the original prompt. They are nearly identical, which suggests that a good approximation to the optimal distortion-rate curve can be obtained by considering pruned prompts only.

A key first step towards extending our algorithm for natural language prompts is the observation that all prompt compression methods in the literature work by *pruning* tokens, i.e., (1) they are non-generative, i.e., work by removing tokens from the original prompt and therefore do not generate any new tokens, and (2) they preserve the order of the tokens as they appear in the input sequence. Hence, to compute the fundamental limit for the schemes that compress the prompt by pruning, it is enough to consider $\mathcal{M}_x$ to be the sequences that are obtained from $x$ by removing some number of (not necessarily contiguous) tokens. Then, we have $|\mathcal{M}_x| = 2^{\text{len}(x)}$, irrespective of the vocabulary size $|\mathcal{V}|$.

In Fig. 8, we observe that the distortion-rate curves obtained by restricting $\mathcal{M}_x$ to be only those sequences obtained from $x$ from via pruning, are nearly identical to the original curves, where we take $\mathcal{M}_x$ to be all shorter sequences. This suggests two things: (1) there is no fundamental drawback to considering compression schemes are not generative, i.e., work by pruning the original prompt, and (2) we can approximate the optimal distortion-rate function reasonably well by considering only pruned versions of the prompt as possible compressed prompts. Thus, in principle, we can replicate experiments with natural language prompts of the same lengths (4 to 10) as the binary prompts in our experiments above, with the same computational cost. However, it is difficult to identify sufficiently rich natural language prompts of such short lengths for which compression is a reasonable problem, and hence, use binary prompts (generated from a Markov chain, to model the dependence between tokens) with natural language queries (since there is no computational restriction on the vocabulary of the queries) to run experiments such as those in Sec. 4.2 and App. F at scale.

Nonetheless, to illustrate that we can get meaningful results by considering such "pruned" compressed prompts, we generate a small natural language dataset using GPT-4 [38], as described in App. F.2.1.

To make sense of the computational complexity involved in computing the optimal distortion-rate curve, consider the following toy example: suppose that the vocabulary size is 10, and that the lengths of all prompts are 100. Then, the number of possible compressed prompts is of the order of $10^{100}$. When restricted to compressed prompts obtained by simply "pruning" the input, this number reduces to nearly $2^{100}$, which is still large. However, since the length of the compressed prompt is at most 100, the number of points on the lower-left convex envelope in Algorithm 1 is at most 100. Hence, the major bottleneck is in computing the $\boldsymbol{D}_x$ quantities, which require $10^{100}$ inference calls. Once these quantities are known, the complexity of Algorithm 1 itself is negligible, even for large vocabulary sizes and prompt lengths. One option is to approximate the optimal curve as best as possible while limiting the number of LLM inference calls made; we provide preliminary results in this direction in App. G.3. To exactly compute the optimal curve while avoiding the $10^{100}$ inference calls requires some assumptions to be made about $\phi_{\text{LLM}}$ and some structure on the generated outputs; we leave this for future work.

# F   Experiment details

## F.1   Synthetic data experiments

We provide additional details regarding our synthetic dataset and how we fine-tuned all models. Experiments were run on three different machines, two of which are identical machines with an AMD Ryzen Threadripper PRO 5975WX CPU (32 cores), 256 GB of system RAM, and 2x Nvidia RTX 4090 GPUs with 24 GB each. We also ran experiments on a DGX machine with an AMD EPYC 7742 64-Core Processor, 512 GB of system RAM, and 4x 80GB SMX4 A100 GPUs. The duration of LLM fine-tuning on the synthetic dataset varies, depending on the model being fine-tuned. In general, it takes 10 to 30 minutes for a single fine-tuning run. Running the code necessary to reproduce all plots takes several hours.

We use code from the LLMLingua and Selective Context GitHub repos, which are released under the MIT license. In our experiments, we use the following models: Mistral 7B Instruct v0.2 (Apache-2.0), RoBERTa Base (MIT), and Pythia 1B deduped (Apache-2.0).

Each method requires a rate or threshold parameter $r$, for which we use $r \in \{0.04, 0.1, 0.2, 0.3, 0.4, 0.5, 0.6, 0.7, 0.8, 0.9, 0.96, 0.99, 1.0\}$ in our experiments. However, the length of the returned compressed prompt might not be faithful to this rate parameter, so for each $r$, we report the average rate and average distortion on the examples in our validation dataset. LLMLingua and LLMLingua Query have one additional parameter controlling the size of each segment the prompt is broken into before compressing each segment. We found that using a segment size of 2 works best for our synthetic dataset.

## F.2   Natural language experiments

We train our models on the same DGX system used in the synthetic dataset experiments. Unlike the synthetic dataset experiments, however, we train these XLM RoBERTa Large (MIT license) models with the single precision (float32) data format and use full fine-tuning rather than LoRA. We observed non-negligible performance improvements with this configuration over bfloat16 with LoRA. As a result, training took one to two hours and 30 GB of VRAM on a single A100. We used the same set or rate and threshold parameters as done in the synthetic data experiments.

### F.2.1   Small natural language dataset

Solving for the optimal rate-distortion curve requires a substantial amount of compute as mentioned in App. E. To overcome this, we construct a synthetic dataset consisting of binary string prompts, natural language queries, and numerical and yes/no answers. We show a few examples of the validation partition of our synthetic dataset in Table 2.

Table 2: One example of each query from the validation set of our synthetic dataset

| Prompt | Query | Answer |
|--------|-------|--------|
| 110011111 | Count the number of 1s. | 7 |
| 11111 | Count the number of 0s. | 0 |
| 00000111 | Compute the parity. | 1 |
| 11011111 | What is the length of the longest subsequence of 0s or 1s? | 5 |
| 0110 | Is the binary string a palindrome? | Yes |
| 1100111100 | Count the number of transitions from 0 to 1 and 1 to 0. | 3 |
| 111111 | Predict the next bit. | 1 |

We curated a small natural language dataset to generate results shown in Fig. 5 by prompting GPT-4 [38] to provide short natural language prompts of 15 tokens or less, provide four questions about each prompt, and give the answer. Afterward, we modified some of the questions and prompts slightly when the generated prompt by GPT-4 was too long or the questions and answers contained too much overlap with each other for a given prompt. In total, our dataset consists of ten prompts with four questions each. A few examples of our dataset are shown in Table 3.

Table 3: One example of each prompt from our natural language dataset.

| Prompt | Query | Answer |
|---|---|---|
| After dinner, the cat chased a mouse around the house. | What was the cat doing? | The cat was chasing a mouse. |
| The dog barked loudly at the passing mailman on a quiet street. | Where did the barking occur? | On a quiet street. |
| After school, the child played with toys in the cozy living room. | When was the child playing? | After school. |
| At the art gallery, the artist painted a colorful mural on the wall. | Where was the painting done? | On the wall at the art gallery. |

### F.2.2 Choice of distortion metric

The proper choice of distortion function is important to meaningfully measure the change in performance (distortion) of the LLM as the rate is varied. Ideally, a distortion metric where two texts with the "same meaning," as would be determined by humans, achieve a "low distortion" is the gold standard, but this metric is unknown. This is a crucial open problem not just for our work but also for the fair benchmarking of LLMs in general.

For our results on natural language, we report our results using the following distortion metrics (or rather, "1−" these, since these are similarity metrics and we want a low distortion to mean high similarity). rougeL [23] is a standard metric used to evaluate summaries, which does so by computing the precision and recall between the tokens in the generated text and the reference texts. In contrast, BertScore [45] computes contextualized embeddings for the generated tokens and reference tokens and uses the pairwise cosine similarity among them to produce the final score. The authors of the BertScore work highlight that their metric correlates better with human judgements than all other metrics (rougeL included). Regardless, our results with these two metrics are in agreement with each other, suggesting that a "good enough" metric may be sufficient. A popular approach in current literature is to ask GPT4 to give a score on the similarity between generated and reference texts. Although it has been shown that humans agree with GPT4's evaluation more than the evaluation of other humans [49], we are skeptical of this metric because it is not reproducible, and GPT4 has biases that may result in unfair or inaccurate evaluations. Additionally, our theoretical framework is general and does not assume any specific distortion function. In particular, it can also be used with new distortion functions that better capture semantic notions when they are discovered.

### F.3 LLM fine-tuning

### F.3.1 Synthetic dataset

Given that our synthetic dataset of binary prompts is not naturally in the distribution of training data of LLMs, we use Mistral 7B Instruct v0.2 [39] as our black-box model, and fine-tune it on tuples of (prompt, query, answer). This is also known as "instruction fine-tuning;" we only compute the loss over the answer.

Each prompt compression method requires an LLM as part of its compression algorithm; we fine-tune Pythia 1B deduplicated [40] for Selective Context and LLMLingua-based methods. Selective Context and LLMLingua only use negative log-likelihood scores over the prompt, so for these methods we fine-tune with the next-word prediction over the prompts. For LLMLingua Query, we place the (query, prompt, answer) tuple into context and then perform next token prediction over the entire context. We place only the query and prompt into the context for the prompt compression step (inference time).

LLMLingua-2 methods require an additional label set for every prompt as ground-truth answers to teach the model to predict which tokens should be kept. For our dataset, gathering the labels for each prompt is deterministic if the query is known, so it is easy to assemble the label set for query-aware LLMLingua-2 methods. For example, for the query "Is the binary string a palindrome?" we can easily choose the shortest sequence of tokens from the input that is also a palindrome (if the answer is "yes") as the ground-truth compressed prompt. For QuerySelect and Adaptive QuerySelect, both of which are query-aware, we put the query and prompt into context and then train the LLM to predict which tokens to keep using the constructed label set. This process is less straightforward for query-

agnostic LLMLingua-2 since it is not clear how to assign the labels without the query. In this case, we choose the ground-truth compressed prompt to consist of the highest entropy tokens. Given the Markov chain from which our prompts were generated, these tokens contain the transitions between bits. For all LLMLingua-2 methods, we fine-tune RoBERTa Base [41].

We conduct a grid search over a set of hyperparameters before fine-tuning the final model used for each method. Specifically, we use the training set to fine-tune a model with all combinations of hyperparameters, evaluate the final performance on each model with a test set, and choose the combination of hyperparameters leading to the best performance. We then merge the train and test set and train with the chosen hyperparameters and do a final evaluation on the validation, which is the dataset used in the results of this paper.

All models are searched over the same learning rate $\{5e-6, 1e-5, 5e-5, 1e-4\}$, batch size $\{16, 32\}$, LoRA rank $\{16, 32, 64, 128\}$, and LoRA alpha $\{16, 32, 64, 128\}$ hyperparameters. For the number of training epochs, we search over $\{1, 2, 4\}$ for Mistral 7B Instruct v0.2 and Pythia 1B deduplicated, and $\{8, 12\}$ for RoBERTa Base.

We report our final set of hyperparameters used to fine-tune the LLM used for each prompt compression method in Table 4.

Table 4: Final set of hyperparameters used to train the LLM used in each prompt compression method.

| Method | Tokenization | Epochs | Batch Size | Learning Rate | LoRA Rank | LoRA Alpha |
|---|---|---|---|---|---|---|
| Selective Context | Standard | 1 | 16 | 5e-5 | 32 | 32 |
| Selective Context | Forced | 1 | 16 | 5e-5 | 128 | 64 |
| LLMLingua | Standard | 1 | 16 | 5e-5 | 32 | 32 |
| LLMLingua | Forced | 1 | 16 | 5e-5 | 128 | 64 |
| LLMLingua Query | Standard | 4 | 32 | 1e-4 | 128 | 128 |
| LLMLingua Query | Forced | 4 | 16 | 1e-4 | 64 | 128 |
| LLMLingua-2 | Standard | 12 | 32 | 1e-4 | 128 | 128 |
| LLMLingua-2 | Forced | 12 | 32 | 1e-4 | 64 | 128 |
| QuerySelect | Standard | 12 | 32 | 1e-4 | 128 | 128 |
| QuerySelect | Forced | 12 | 32 | 1e-4 | 64 | 128 |
| Adaptive QuerySelect | Standard | 12 | 32 | 1e-4 | 128 | 128 |
| Adaptive QuerySelect | Forced | 12 | 32 | 1e-4 | 64 | 128 |
| Black-box target LLM | Standard | 4 | 16 | 5e-5 | 16 | 16 |
| Black-box target LLM | Forced | 4 | 16 | 5e-6 | 16 | 64 |

### F.3.2 Natural language dataset

We use fine-tuned models available on Hugging Face for the Selective Context, LLMLingua, LLMLingua Query, and LLMLingua-2 prompt compression methods; only QuerySelect and Adaptive QuerySelect, our novel methods, require specialized fine-tuning starting from a pretrained XLM RoBERTa Large [41] model. To fine-tune these models, we modify the LLMLingua-2 training code available in the LLMLingua GitHub repository to accept (prompt, query) pairs as input and classify which tokens of the prompt should be kept. Since the query uses additional context, and since the

max context window of XLM RoBERTA Large is 512 tokens, we shrink the window size dedicated to the prompt to 384 tokens. We used the same hyperparameters used in the LLMLingua-2 paper [14] for fine-tuning, i.e., learning rate 1e−5, batch size 10, epochs 10, and the Adam optimizer with $\beta_1 = 0.9, \beta_2 = 0.999, \epsilon =$1e−8, and weight decay 0.

The dataset used to train our models is a filtered version of Databricks Dolly 15k [44]. Our final filtered dataset contains samples that meet the following conditions: (1) the context is non-empty, (2) the context is between 50 and 5000 characters in length, and (3) the instruction has fewer than 360 characters. After filtering, the dataset contains 4.35k samples. The contexts of this dataset are chunked into windows of 384 tokens, and we prompt GPT-4 to compress the context by asking it only to keep the tokens necessary for responding to the provided instruction. This allows us to construct a label set for the token classification problem of choosing which tokens to keep or remove. Our dataset used for training consists of the instructions (queries), chunked contexts (prompts), and the set of labels for the contexts. 95% of the samples in this dataset are used in the training dataset, and the remaining 5% form the validation set. Table 5 shows the prompt we used for GPT-4, which is a modified version of the prompt used in the LLMLingua-2 paper. All of our code used for constructing the training dataset and training the models are modified from the LLMLingua GitHub repo. Our modified code is available in the code release.

Table 5: Our prompt for GPT-4 to determine which tokens to remove. This prompt was used to construct the dataset for training QuerySelect and Adaptive QuerySelect on natural language. This is a slight modification from the prompt used in the LLMLingua-2 work [14].

| System Prompt | You are an excellent linguist and very good at compressing passages into short expressions by removing unimportant words, while retaining as much information as possible. |
|---|---|
| User Prompt | Compress some text to short expressions, such that you (GPT-4) can answer the query based on the compressed text. Unlike the usual text compression, I need you to comply with the 5 conditions below: |
| | 1. You can ONLY remove unimportant words. |
| | 2. Do not change the order of words. |
| | 3. Do not change the original words, e.g., 'asking' → 'ask' is NOT OK; 'current' → 'now' is NOT OK. |
| | 4. Do not use abbreviations or emojis, e.g., 'without' → 'w/o' is NOT OK; 'as soon as possible' → 'ASAP' is NOT OK. |
| | 5. Do not add new words or symbols, this is very important. For example, 'dedicate 3 hours to each chapter' → '3 hours/chapter' is NOT OK because you add new token '/', just compress it into '3 hours each chapter'. '30 eggs plus 20 eggs equals 50 eggs' → '30+20=50' is also NOT OK because you add new symbols + and =; just compress it into '30 plus 20 equals 50'. |
| | Compress the origin aggressively by removing words only. Please output the compressed text directly. Compress the origin as short as you can, while retaining ONLY the information needed to answer the following query: {query}. |
| | If you understand, please compress the following text: {text_to_compress} |
| | The compressed text is: |

# G Additional experimental results

## G.1 Small-scale datasets

We provide the remainder of our empirical results below. Fig. 9 is similar to Fig. 1, but shows the result when the prompt compression method uses standard tokenization rather than forced tokenization. Fig. 10 shows a direct comparison between the trade-off for methods using standard and forced

tokenization. Fig. 11, Fig. 12, Fig. 13, and Fig. 14 show the rate-distortion trade-off curves for each of the seven queries in our synthetic dataset. Fig. 11 shows forced tokenization with 0/1 loss, Fig. 12 shows forced tokenization with log loss, Fig. 13 shows standard tokenization with 0/1 loss, and Fig. 14 shows standard tokenization with log loss.

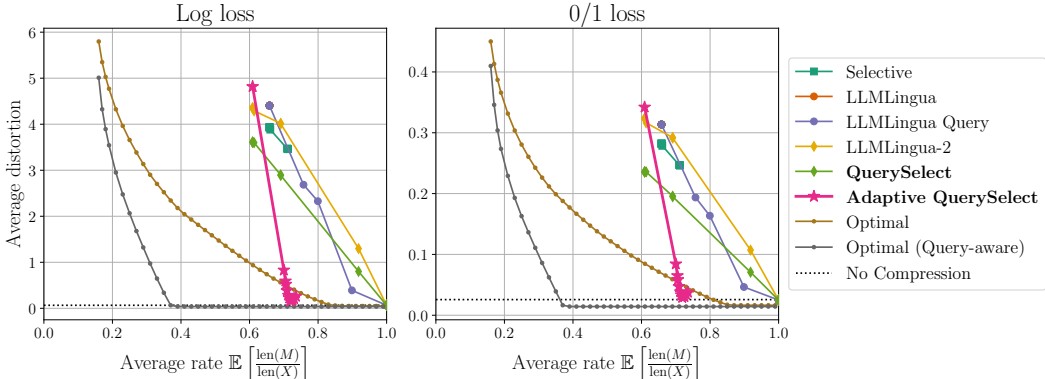

Figure 9: The distortion-rate curves of all prompt compression methods and the optimal strategy attained by solving our dual LP formulation when standard tokenization is used for the prompt. All methods are compared with the log loss **(left)** and 0/1 loss **(right)** metrics.

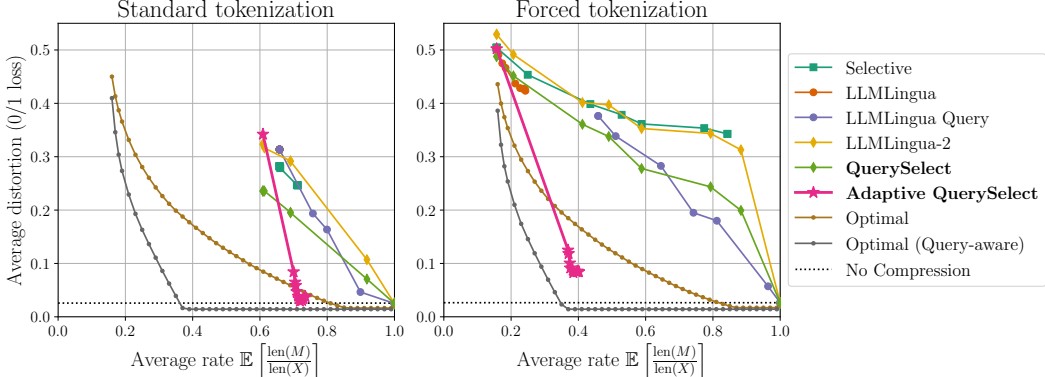

Figure 10: Performance comparison when standard tokenization **(left)** and forced tokenization **(right)** is used on our synthetic dataset. Interestingly, the optimal performance is nearly equivalent between the two, and, for a given rate, methods with standard tokenization match or improve upon the performance of a method that forced separate tokenization of every bit. However, standard tokenization results in compression of 1 to 4 tokens on our dataset, whereas forced tokenization compresses up to 10 tokens, allowing for a greater range of rates.

### G.2   NarrativeQA

We compare existing methods on the NarrativeQA [18] dataset. Specifically, we use the summaries from the dataset as the prompt, and use the query and answer as given. Since the sizes of the prompts are hundreds of tokens and approximately 3,500 samples, it is not feasible to compute the optimal rate-distortion curve for this dataset. Instead, Fig. 16 showcases the performance of our proposed methods over existing methods. In particular, Fig. 16 shows the same conclusion as Fig. 5: our proposed methods are better for rates less than 0.5 and remain competitive for rates about 0.5. Table 6 shows the average time required to compress a prompt on the NarrativeQA dataset and our curated small-scale NLP dataset. Since our methods are adapted from LLMLingua-2, our methods share the same timings.

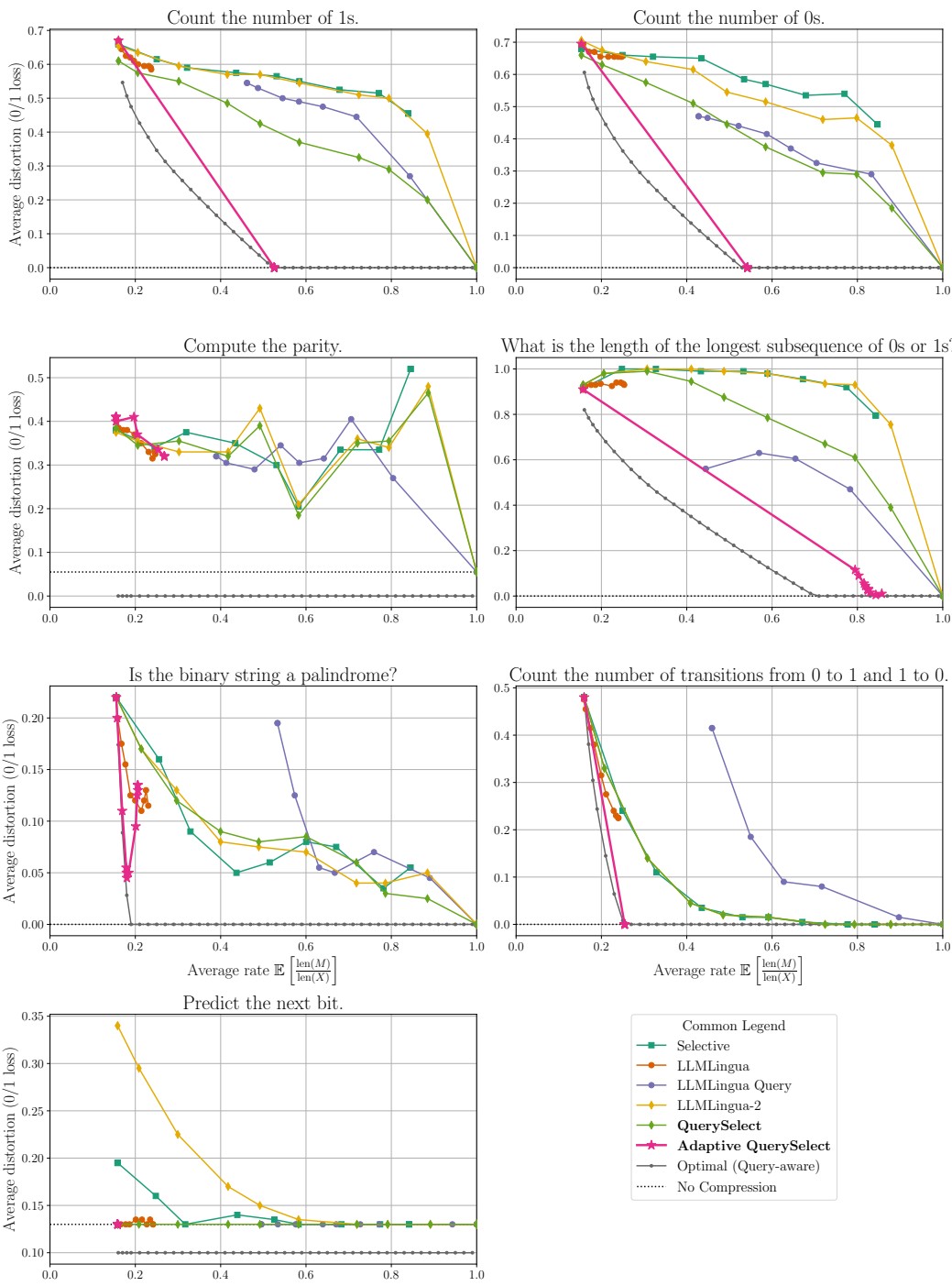

Figure 11: The rate-distortion trade-off of all methods on each individual query for forced tokenization and 0/1 loss.

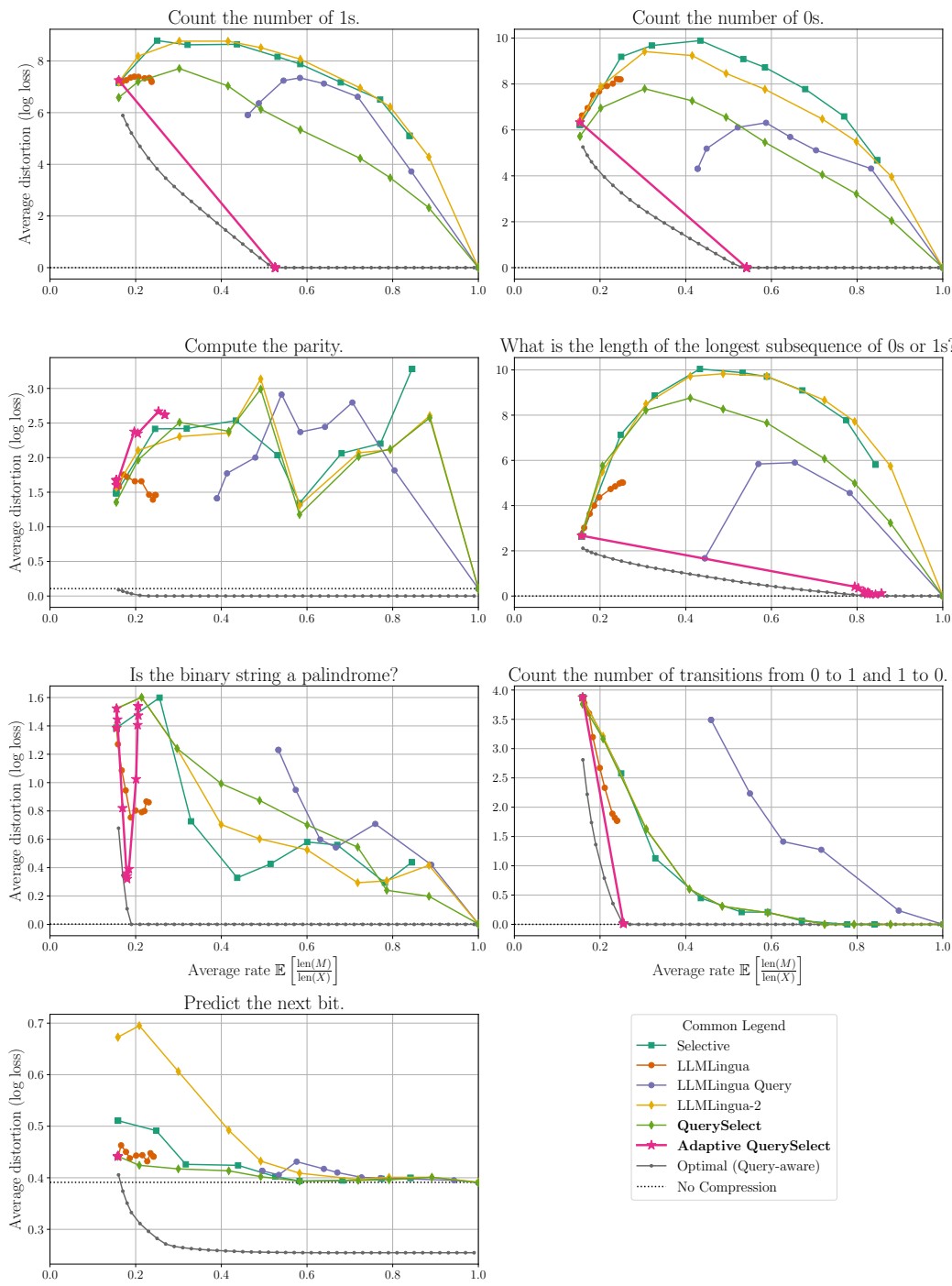

Figure 12: The rate-distortion trade-off of all methods on each individual query for forced tokenization and log loss.

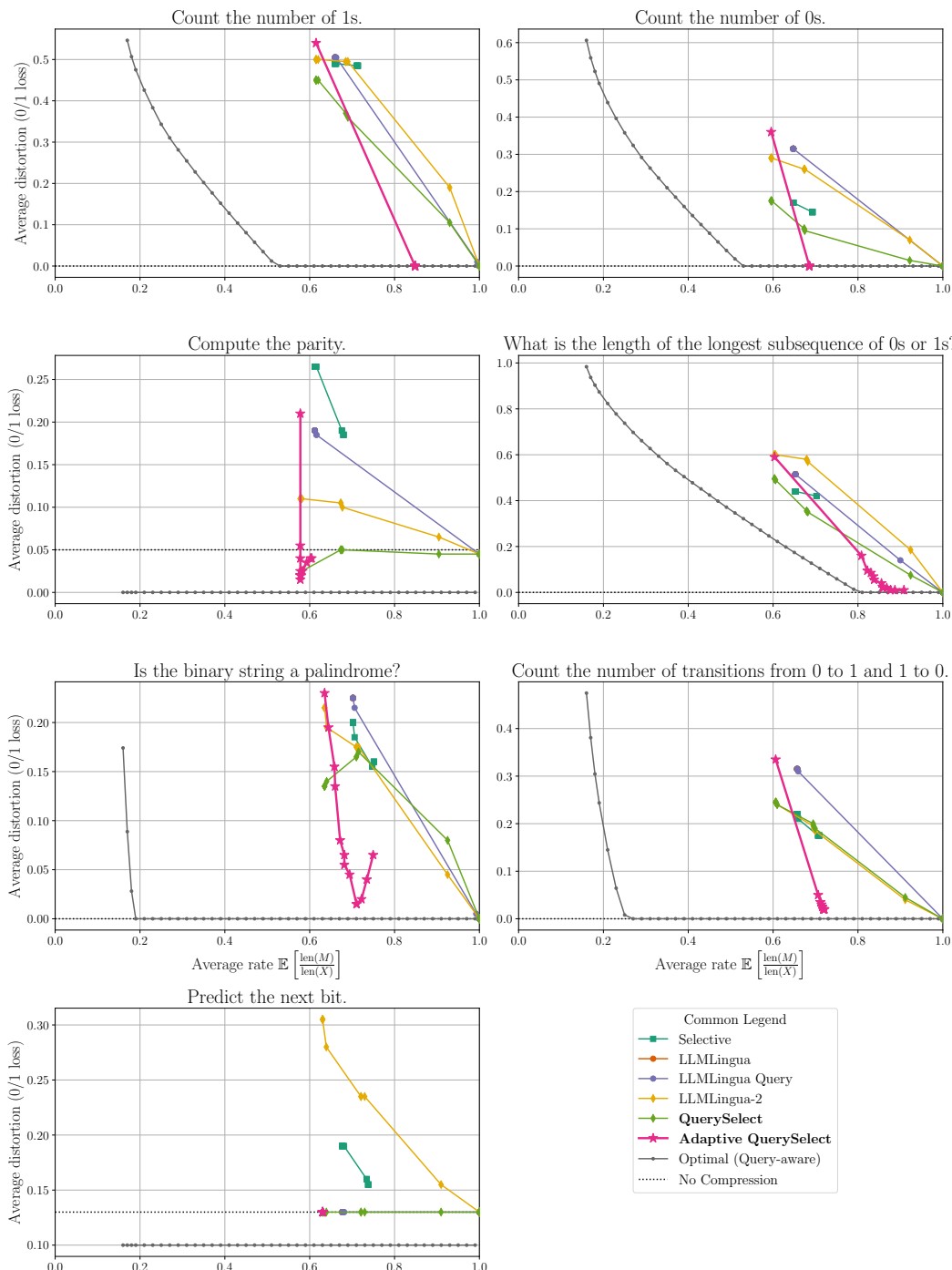

Figure 13: The rate-distortion trade-off of all methods on each individual query for standard tokenization and 0/1 loss.

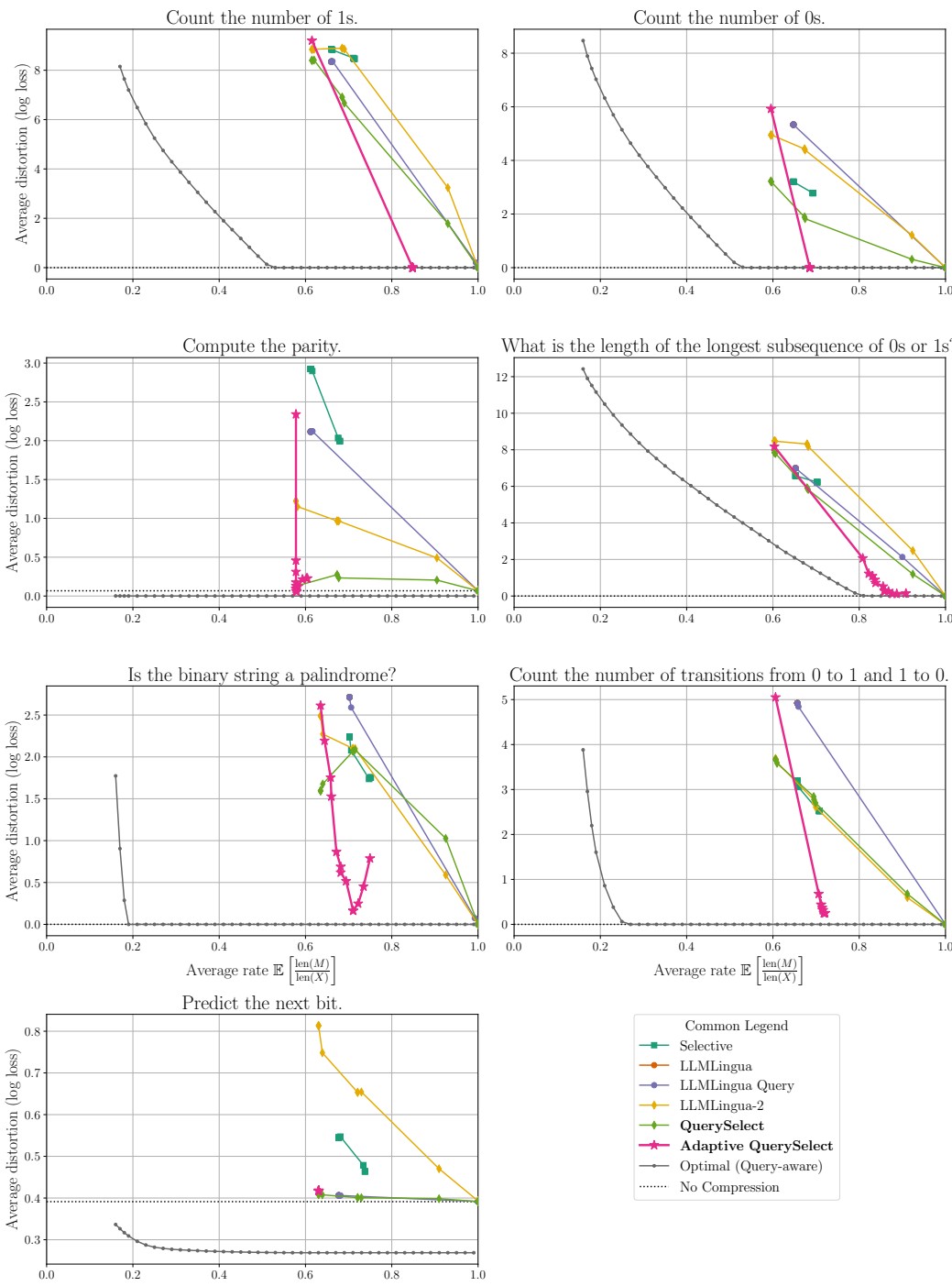

Figure 14: The rate-distortion trade-off of all methods on each individual query for standard tokenization and log loss.

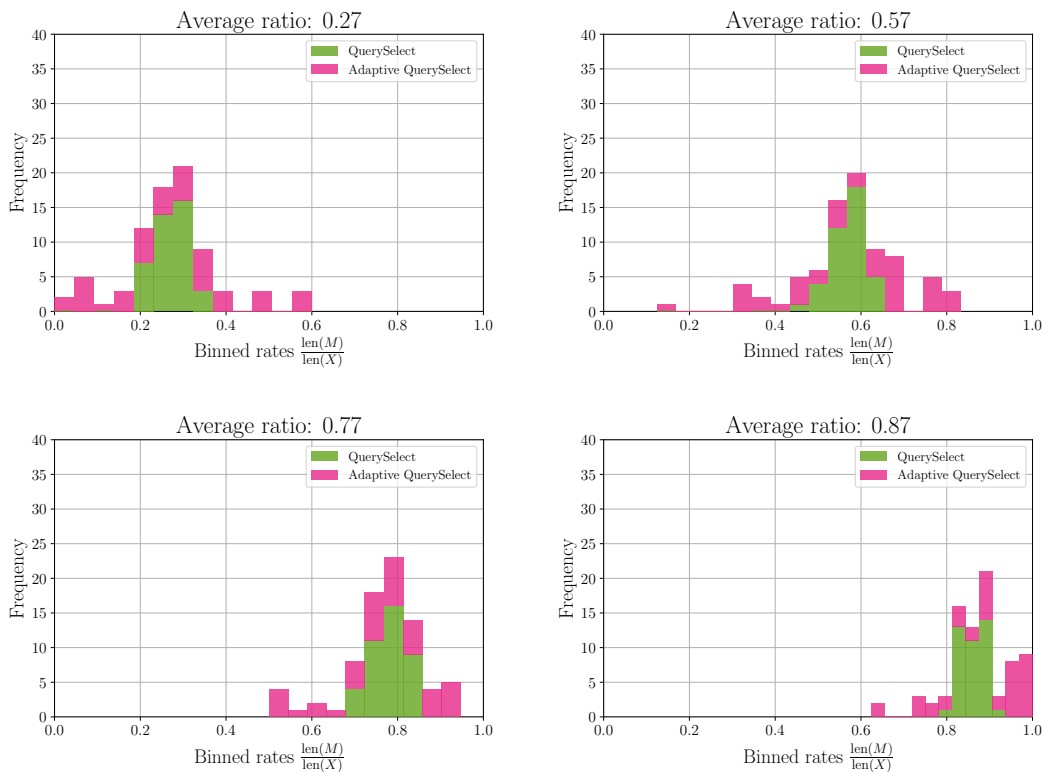

Figure 15: Histograms of the rates for QuerySelect and Adaptive QuerySelect for a given average rate from the left-side figure of Fig. 5. These figures show that Adaptive QuerySelect has a larger spread of rates across the samples of the natural language dataset. In particular, Adaptive QuerySelect has greater flexibility in choosing an appropriate rate for a given (prompt, query) pair.

Table 6: Average time required to compress a single prompt (seconds).

| Method | NarrativeQA | Small NLP Dataset |
|---|---|---|
| Selective | 1.043 | 0.049 |
| LLMLingua | 0.510 | 0.273 |
| LLMLingua Query | 1.060 | 0.530 |
| LLMLingua-2 | 0.113 | 0.044 |
| QuerySelect | 0.114 | 0.043 |
| Adaptive QuerySelect | 0.114 | 0.043 |

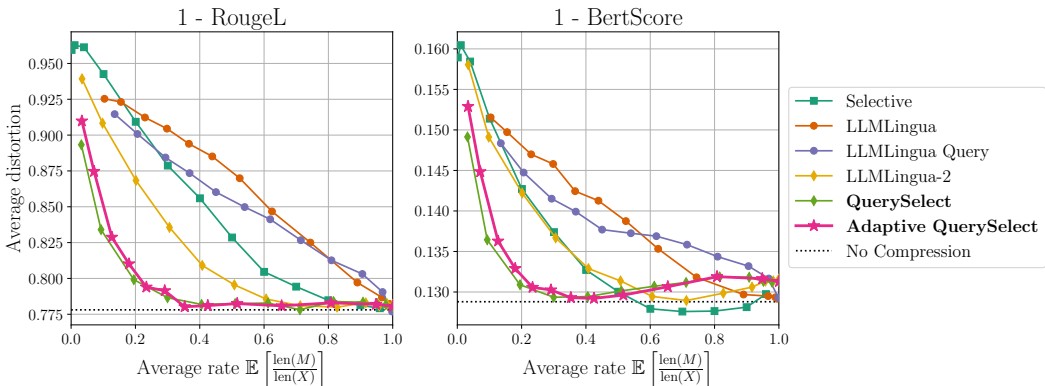

Figure 16: Comparison among all prompt compression methods on the NarrativeQA [18] summaries dataset. We show the rate-distortion trade-off for RougeL [23] **(left)** and BertScore [45] **(right)**. Since a higher RougeL and BertScore metric is better, we plot "$1-$ the computed average distortion" so that a higher rate should yield a lower loss.

### G.3  NarrativeQA Beam Search

As discussed in the Conclusion and App. E, computing the optimal rate-distortion curves for large-scale datasets, where the prompts consist of hundreds or thousands of tokens, is intractable. The difficulty lies in computing $\boldsymbol{D}_x$, which requires an inference call for every possible compressed prompt for a given prompt. Even for our curated small-scale dataset, whose largest prompt consists of 15 tokens, it is not feasible to compute the "true" optimal curve outright. Instead, we considered the set of compressed prompts constructed from in-place token removal from the original prompt, which significantly reduced the number of inference calls in constructing $\boldsymbol{D}_x$ from $\sum_{i=1}^{\text{len}(x)-1} |\mathcal{V}|^i$ to $2^{\text{len}(x)}$, where $|\mathcal{V}|$ is the size of vocabulary of the tokenizer and $x \in \mathcal{X}$ is the prompt. While this approach does not yield the "true" optimal curve, it does provide an optimal curve to all existing prompt compression methods since they all strategically remove tokens in place. Thus, we can still establish the fundamental limit of existing methods!

However, when $\text{len}(x)$ is a few hundred or thousand, as is the case for large-scale datasets, $2^{\text{len}(x)}$ is far too many inference calls to make $\boldsymbol{D}_x$ to be practical. Ideally, one can approximate the optimal curve by finding an upper bound; to do this, we use beam search. Our search space is over all $2^{\text{len}(x)}$ binary masks (a binary mask, when applied to the prompt, forms a compressed prompt). Each mask is assigned a distortion value by computing the distortion between the ground truth answer and the generated answer when the black-box LLM is given the mask's associated compressed prompt and the query. Thus, we can construct the search tree over the binary masks and use beam search to find masks with low distortion.

The search tree begins with the "all ones" mask at the root node ($l = 0$); each of the root node's children ($l = 1$) contains a bit flipped to 0 in precisely one of the $\text{len}(x)$ positions. At the third level ($l = 2$), each node inherits a 0 at the same position as its parent and then flips a 1 to a 0 so that each mask at $l = 2$ has precisely two 0s. This pattern continues throughout the rest of the tree, resulting in masks with $l$ 0s at level $l$. Furthermore, the branching factor $F_l$ of our beam search is the number of children nodes a given node has at level $l$, is $N - l$. At level $l$, accounting for all nodes that are shared among the parent nodes in level $l - 1$, there are $\binom{N}{l}$ nodes, where $N = \text{len}(x)$. As a sanity check, this tree contains $\sum_{k=0}^{N} \binom{N}{k} = 2^{\text{len}(x)}$ nodes in this tree, so all binary masks of length $N$ are in the tree. We can search every node in the tree as long as the beam width $B$ is large enough to contain every node at the broadest level of the tree. To achieve this, we need to set $B = \max_l \binom{N}{l} = \binom{N}{\lfloor N/2 \rfloor}$.

Recall that our purpose for using beam search is to drastically reduce the number of LLM inference calls required for constructing $\boldsymbol{D}_x$ while retaining a competitive upper bound to the optimal curve. The cost $C$ associated with the beam search, which is the number of nodes visited in the tree (i.e., the number of LLM inference calls) is

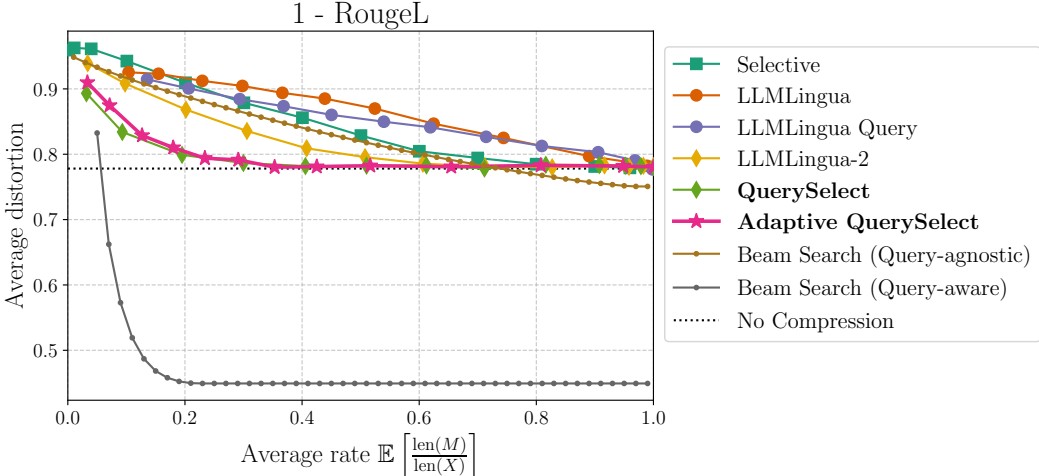

Figure 17: Comparison between existing prompt compression methods (replicated here from Fig. 16) and our approximation to the optimal rate-distortion curves via beam search on NarrativeQA.

$$C = B \sum_{l=0}^{N} F_l = B \sum_{l=0}^{N} (N - l) = B \sum_{k=0}^{N} k = \frac{BN(N+1)}{2}. \tag{10}$$

Note that, in this case, we search over $C < |\mathcal{M}_x|$ in (LP) compressed prompts, so beam search will provide an upper bound. When $N$ is large, checking $\mathcal{O}(BN^2)$ nodes is a drastic reduction over checking $2^N$ nodes, but the growth rate is still too large. For example, running beam search on a single prompt of just 100 tokens will require on the order of $10^6$ LLM inference calls, or roughly 11.5 days if assuming a single inference call takes one second. To further reduce the number of checked nodes, we can chunk the binary masks into spans of bits, and flip entire spans from 1 to 0 at each level. With this approach, we can effectively control the length of the binary masks we search over, which we call $N_{\text{eff}}$, for a given budget/cost $C$. Solving (10) for $N$, we arrive at

$$N_{\text{eff}} = \frac{-1 + \sqrt{1 + \frac{8C}{B}}}{2}$$

This approach allows us to choose the total number of LLM inference calls $C$ per prompt that we will spend on beam search for a particular beam width $B$. Since we are chunking the binary mask into spans and then masking the spans as we search, we have reduced the search space and the resulting tree will be smaller than the $N_{\text{eff}} = N$ case.

Fig. 17 compares existing methods to our beam search upper bounds for $C = 4000$ and $B = 5$. Although LLMLingua-2-based methods can outperform the upper bound in the query-agnostic case for most rates, the query-agnostic beam search approach shows room for improvement from existing methods. Impressively, the gap between existing methods and the query-aware case is quite large, even for the methods that use the query to compress the prompt.

