# OpenReview forum: "Fundamental Limits of Prompt Compression: A Rate-Distortion Framework for Black-Box Language Models"
_NeurIPS.cc/2024/Conference — NeurIPS 2024 poster_

### Official Review · Reviewer_sxbS · 2024-07-17

**Soundness:** 3
**Presentation:** 4
**Contribution:** 3
**Rating:** 7
**Confidence:** 3

**Summary:**

The paper formulates hard prompt compression as a rate-distortion problem. An algorithm is provided for hard prompt compression, as well as another to estimate the RD curve (Algorithm 1). Experiments on synthetic as well as benchmark datasets are provided with comparisons to previous methods.

Note:
- I'm more than willing to change my score significantly if the questions are addressed appropriately.
- I don't know much about LLMs.
- I'm not familiar with previous literature on prompt-compression.
- I'm familiar with convex optimization, but it's hard to evaluate the correctness or novelty of Algorithm 1.
- I'm familiar with info theory in general.

**Strengths:**

- Overall the paper is very well written. I was able to follow everything to a reasonable degree even without knowing much about LLMs.
- The paper takes a principled approach, through the lens of info theory, to prompt-compression to investigate fundamental limits.
- The overall conclusion points towards a space for improvement in current methods (summarized in Figure 1.), with the caveats mentioned in the questions/weakness.

**Weaknesses:**

- The synthetic data is very limited as it has a binary support (although this is highlighted in as a limitation in the appendix).
- The gold standard for the distortion functions, as far as I understand, would be to have a distortion function that is low when two sentences have "the same meaning", as measured by humans, and high otherwise. Of course, this is function doesn't exist (as far as I know). It isn't obvious why the chosen distortion metrics (i.e., log-loss and 0/1) capture "semantics" as originally intended by the authors.

Smaller issues
- Line 123: notation for a pair of objects inconsistent with the previous parts of the text (i.e., used $[M, Q]$ instead of $(M, Q)$)
- Line 147: typo, "taken with respect" -> "taken with respect to"
- Line 158: typo, "The dimensions of problem" -> "The dimensions of the problem"

Suggestions (this is conditioned on there being reasonable explanations for the questions in the Questions section):
- I would suggest focusing the paper more about the empirical results and why the RD formulation makes sense for this problem (the ML community is not too familiar with info theory, so the formulation itself is a contribution in my opinion).
- The entirety of section 3.3 is not too important for the story and could be entirely moved to the appendix. Instead, replace 3.3 with a discussion as to why this RD formulation captures the "rate-semantics" trade-off originally intended by the authors, by providing evidence that the distortion functions proposed can actually measure semantic distortion.

**Questions:**

> Line 261: "taking $P_{XQY}$ to be the empirical distribution on the dataset"

If the empirical distribution is used in the RD formulation, isn't this just overfitting to the sample?

> Line 108 and Equation (1)

I don't understand why the log-loss, or 0/1 loss, under the LLM is a reasonable distortion function to measure semantic distortion. Please see "Weakness" section.

**Limitations:**

There's a large limitations section in the appendix which I appreciate.

---

> ### Author Rebuttal · Authors · 2024-08-07
>
> We thank the reviewer for the careful reading of the paper and constructive suggestions and feedback. We provide our response to the reviewer’s individual comments here, and strongly encourage the reviewer to check our global response for updates regarding our new natural language experiments, which is also relevant to the reviewer’s comments. Please find our response to the questions below:
>
> * **Limited dataset**: We include two new experiments on natural language data in the global response.
> * **Choice of distortion metric**:  Indeed, the proper choice of distortion function is important to meaningfully measure the change in performance (distortion) of the LLM as the rate is varied. Ideally, a distortion metric as described by the reviewer (“low distortion” for sentences with the “same meaning,” as measured by humans) is the gold standard, but this metric is not known. This is very much a crucial open problem for not just our work, but for fair benchmarking of LLMs in general.
>
>     * **0/1 loss for synthetic dataset**: For our synthetic dataset consisting of binary strings, the 0/1 loss is the best. This is because (1) the answers are numerical or binary answers, so there is no notion of semantic relatedness between two answers, and (2) the answers are single tokens, so the most practical thing is to check for an exact match between the answers. We also include the log loss metric to include some diversity in our evaluations. Although the structure of the curves are different, the relative ordering of the curves and their trends are very similar to the curves in the 0/1 loss plot.
>
>     * **rougeL and BertScore: Semantic distortion metric for natural language datasets**: For our new results on natural language, we report our results using these (or rather, “$1-$” these, since these are similarity metrics and we want a low distortion to mean high similarity):. rougeL is a common metric used to evaluate summaries, which does so by computing the precision and recall of the tokens in the generated and reference texts, while BertScore computes contextualized embeddings for the tokens and uses the pairwise cosine similarity among them to produce the final score. The authors of the BertScore work highlight that their metric correlates better with human judgements than all other metrics (rougeL included) [1]. Regardless, our results with these two metrics are in agreement with each other, suggesting that a “good enough” metric may be sufficient. A popular approach in current literature is to ask GPT4 to give a score on the similarity between generated and reference texts. Although it has been shown that humans agree with GPT4’s evaluation more than the evaluation of other humans [2], we are skeptical of this metric because it is not reproducible and GPT4 has biases which may result in unfair or inaccurate evaluations. Additionally, we would also like to mention that our theoretical framework is general and does not assume any specific distortion function. In particular, it can also be used with new distortion functions that better capture semantic notions, when they are discovered in the future.
>
> * **Discussion of rate-semantic trade-offs in Section 3.3**:
>     * We thank the reviewer for this suggestion. We will add a discussion on the semantic metrics (as in the answer above) together with our NLP experimental results in the revised manuscript.
>     * We maintain that Section 3.3 is important because our primal and dual LP formulations, particularly Algorithm 1, not only form the basis of our theoretical contributions by allowing us to compute the optimal curve efficiently but also demonstrate that our theoretical results hold for any distortion metric, including those designed to capture distortion in the semantic space.
>
> * **Re: If the empirical distribution is used in the RD formulation, isn't this just overfitting to the sample?**: Firstly, we would like to mention that our theoretical framework allows for any choice of the distribution, and Algorithm 1 can still be used to compute the optimal curve with that choice of the distribution.
>
>     * We choose the empirical distribution since it is the most natural choice when the distribution is unknown and we are only given a dataset. This is also how other works computing information-theoretic limits of compression in various settings approximate the underlying distribution ([33, 34] in our paper).
>
>     * Alternatively, one might assume a parametric distribution and try to learn the parameters from the dataset. However, it is unclear what the right parametric model for text (particularly LLM prompts) should be.
>
>      * It is true that a very small number of samples (i.e., when the dataset is not a good representation of the true distribution) can lead to an optimal compressor that overfits. In our experiments on synthetic prompts, we observe that LLMLingua-2 Dynamic nearly matches the optimal curve for some rates. Had the optimal compressor significantly overfit to the dataset, optimality would not have been achieved (although this is not a necessary condition). This suggests that with a sufficiently large dataset, the empirical distribution is a good enough approximation.
>
>     * Finally, in each of our plots, we compare all compression schemes to the optimal curves by evaluating them on the same dataset, so the comparison is fair.
>
>     This is indeed an interesting problem, and we will add a discussion highlighting the choice of distribution in the revised manuscript.
> ___
> **References**:
>
> [1] Tianyi Zhang*, Varsha Kishore*, Felix Wu*, Kilian Q. Weinberger, & Yoav Artzi (2020). BERTScore: Evaluating Text Generation with BERT. In International Conference on Learning Representations.
>
> [2] Rafael Rafailov, Archit Sharma, Eric Mitchell, Christopher D Manning, Stefano Ermon, & Chelsea Finn (2023). Direct Preference Optimization: Your Language Model is Secretly a Reward Model. In Thirty-seventh Conference on Neural Information Processing Systems.

---

> > ### Comment · Reviewer_sxbS · 2024-08-09
> >
> > The added discussion on "semantic distortion" is very satisfying. I'm happy with the response and have updated my score accordingly.

---

> > > ### Author Response · Authors · 2024-08-09
> > >
> > > We thank the reviewer for their timely response, and are happy that the reviewer is satisfied with our rebuttal. We further extend our gratitude to the reviewer for increasing their score.
> > >
> > > As per the reviewer's suggestion, we will include a discussion on semantic distortion in the revised paper.

---

### Official Review · Reviewer_Jydp · 2024-07-17

**Soundness:** 3
**Presentation:** 3
**Contribution:** 3
**Rating:** 7
**Confidence:** 4

**Summary:**

The paper studies the distortion-rate function for prompt compression and proposes an linear programming based algorithm that produces compressed "hard" prompts and is suitable for black box models. The authors provide empirical results on a synthetic dataset illustrating a gap between the existing prompt compression approaches and the theoretically optimal distortion-rate tradeoff and show that the proposed algorithm outperforms the existing methods.

**Strengths:**

- The paper provides sufficient detail on related work and motivates the problem well.

- The proposed distortion-rate formulation is simple yet easy to understand/interpret. It also allows for query-adaptive solutions with simple modifications to the formulation.

- The empirical results on the synthetic dataset demonstrate the improvements over existing prompt compression methods, which are provably (since the dataset is synthetic) worse than the optimal compressor.

**Weaknesses:**

- It would be great to see some experimental results with real datasets.

- What is the reason that the rate is normalized but the distortion is not? Have the authors tried different combinations?

- How does the efficiency of the proposed method compare with existing prompt compression methods? Did the authors compare how long each method takes to compress the same prompt?

**Questions:**

In addition to the questions above, out of curiosity, do the authors expect any challenges when the compressor is conditioned on multiple queries?

**Limitations:**

The authors mention limitations in the last section.

---

> ### Author Rebuttal · Authors · 2024-08-07
>
> We thank the reviewer for the comments and constructive feedback. We provide our response to the reviewer’s individual comments here, and strongly encourage the reviewer to check our global response for updates regarding our new natural language experiments, which is also relevant to the reviewer’s comments. Please find our response to the questions below:
>
> * **Real (natural language) datasets**:  We include two new experiments on natural language data in the global response.
>
> * **Normalization**: Normalizing the distortion in the same sense as the rate could, in theory, be done by defining the distortion to be $\mathbb{E}\left[ \frac{\mathsf{d}(Y, \phi_{\text{LLM}}(M,Q))}{\mathsf{d}(Y, \phi_{\text{LLM}}(X,Q))}\right]$. However, this is not reasonable in practice since $\mathsf{d}(Y, \phi_{\text{LLM}}(X,Q))$ is usually small (in fact, they are often 0 for the 0/1 loss plots), leading to very large normalized distortion values. A different way to normalize would be to subtract the distortion of the uncompressed prompt, but this would only shift the plots such that the line corresponding to “No Compression” in our plots is at 0. This provides less information than our plots, since we then no longer know how much the “No Compression” distortion is.
> One might also ask whether it is possible to use the rates without normalization, i.e., just using $\mathbb{E}[\mathrm{len}(M)]$ or $\frac{\mathbb{E}[\mathrm{len}(M)] }{ \mathbb{E}[\mathrm{len}(X)] }$ instead of $\mathbb{E}\left[\frac{\mathrm{len}(M)}{\mathrm{len}(X)}\right]$. None of these choices are right or wrong, as they capture slightly different nuances of compression. Choosing the appropriate normalization for a given scenario is certainly an important problem, as pointed out by the reviewer. The key point here is that all these cases can be handled by our framework, by simply changing the associated definitions of rate and distortion to be normalized or unnormalized as needed. Algorithm 1 can still be used to compute the optimal curve, by appropriately redefining the quantities $\boldsymbol{R}_x$ and $\boldsymbol{D}_x$.
>
> * **Efficiency**: The efficiency of our methods is the same as the efficiency of LLMLingua-2, which is faster than LLMLingua methods and Selective Context. The following shows the average time it takes each method to compress a a prompt from our small natural language dataset:
> | Compression method | Time (seconds) |
> | --------- | -----------: |
> | Selective Context | 0.049 |
> | LLMLingua | 0.273 |
> | LLMLingua Query | 0.530 |
> | LLMLingua-2 | 0.044 |
> | LLMLingua-2 Query | 0.043 |
> | LLMLingua-2 Dynamic | 0.044 |
>
>     We are not able to provide the timings for the NarrativeQA dataset since we had to distribute those experiments across machines with differing hardware (which yields an unfair timing comparison) in order to ensure the results were complete before the rebuttal deadline. We thank the reviewer for this suggestion. We agree that reporting the timings/efficiency of the compression methods is useful, and we will include a table for the timings on NarrativeQA, in addition to our small natural language dataset, in the final revision.
>
> * Additional question, **multiple queries**: As far as computing the optimal curve using Algorithm 1 goes, there is no new difficulty when there are multiple queries to condition on. These can all be clubbed into a new “mega-query” $Q’$, and all of the theory follows, replacing $Q$ by $Q’$.  Naively, this approach may also be used with existing prompt compression methods, where all queries of interest are concatenated together. We expect that this may work well if the compressor is trained to accept multiple queries. Currently, all query-aware compressors are trained for a single query, and we are skeptical if they will work well without additional training.

---

> > ### Comment · Reviewer_Jydp · 2024-08-09
> > **response**
> >
> > I thank the authors for their response; my questions have been answered sufficiently. I maintain my score for acceptance.

---

> > > ### Author Response · Authors · 2024-08-09
> > >
> > > We thank the reviewer for their timely response, and are happy that we were able to sufficiently answer all questions.
> > >
> > > We believe that including the table of timings for the compression methods is a useful benchmark to include, and we will add it to the revised paper.

---

### Official Review · Reviewer_cHot · 2024-07-23

**Soundness:** 4
**Presentation:** 3
**Contribution:** 3
**Rating:** 7
**Confidence:** 5

**Summary:**

This paper proposes an information-theoretic framewok for token-level prompt compression for large language models, where the rate is characterized by the expected ratio between the compressed prompt and the original prompt, and the distortion is a cross-entropy based distortion or an accuracy based distortion. The RD problem is then converted to a LP problem, where optimal solutions are given for both query-aware and query-agnostic settings. An algorithm "LLMLingua-2 Dynamic" is proposed that outperforms other LLMLingua-based algorithms.

**Strengths:**

Token-level semantic compression has been an active area of research. The current paper builts an information-theoretic framework inspired by the rate-distortion trade-off in source coding. Algorithms for solving the RD problems are given. Therefore, the proposed framework will be useful for comparing prompt compression methods in the future.

**Weaknesses:**

The motivation for query-aware compression is not clear. If query is provided, then why not compress the answer directly, so the optimal rate would be a constant? The main content focuses on the query-aware setting (we assume that a query is provided in addition to the compressed prompt during the target LLM inference call), but the theorecital discussions in section 3.3 are on the query-agnostic setting. While the query-aware parallel is provided in the appendix, is there any reason for not focusing on the query-aware setting in the first place?

The experiments are not convincing, in that the prompt lengths in the dataset is too short, and that the dataset only contains 7 queries. In practice, the token-level semantic compression is often used to reduce redundancy of long input. Therefore, this work should consider long prompts such as the NarrativeQA dataset. Moreover, the literature review is not thorough -- some baselines are missing (e.g., [1,2]).

[1] Wingate, David, Mohammad Shoeybi, and Taylor Sorensen. "Prompt compression and contrastive conditioning for controllability and toxicity reduction in language models." arXiv preprint arXiv:2210.03162 (2022).
[2] Fei, Weizhi, et al. "Extending context window of large language models via semantic compression." arXiv preprint arXiv:2312.09571 (2023).

**Questions:**

1. Why is the compression rate defined as the expected ratio between len(M) and len(X)? In information theory, the rate is usually measured by units such as bits.
2. In the query-aware setting, if query is provided, then why not compress the answer directly, so the optimal rate would be a constant?
3. In lines 85-86, you said "we assume that a query is provided in addition to the compressed prompt during the target LLM inference call", but later in lines 136-137, you then said "To simplify the presentation, we restrict our discussion here to the query-agnostic setting, and only briefly mention the analogous definitions and results for the query-aware setting." Is there any particular reason for this conflict?
4. In Proposition 1, what do you mean by $\mathbb{R}_+^{\mathcal{M}_x}?$ $\mathcal{M}_x\subseteq \mathcal{V}^*$ right?
5. The proof of Proposition 1 is unclear. In (LP), constants $D_x, R_x$ are used, but only $D_{x,m}, R_{x,m}$ are defined.
6. From the experimental results, it seems that the proposed method can bot achieve high compression rate/low distortion (lossless). Could you discuss this in your paper?
7. Could you provide comparisons to other baselines such as [1] and [2]?
8. Could you conduct experiments on other public datasets, such as the NarrativeQA dataset?

**Limitations:**

Yes.

---

> ### Author Rebuttal · Authors · 2024-08-07
>
> We thank the reviewer for the constructive feedback. We provide our response to the individual comments here, and strongly encourage the reviewer to check our global response for updates regarding our new natural language experiments, which is also relevant to the reviewer’s comments. We also believe there is a misunderstanding regarding the meaning of query-aware prompt compression, which we address in our responses to Q2 and Q3.
> * Q1 (**Unit of rate**): We define the rate to be len(m)/len(x) mainly because this is the convention that is followed in the prompt compression literature. Both len(m) and len(x) are measured in “tokens”, so the unit of rate is “tokens (of compressed prompt) per token (of prompt)”. In classical information theory, sequences are compressed to binary strings, hence the unit is “bits per source symbol”. Even in information theory, it is indeed the length that is used as the proxy for rate in variable-rate coding [1] and more recent “one-shot” compression setups [2,3].
> * Q2 (**Compression of answer**):  Providing the answer as an input to the LLM might not result in the correct output, e.g. let the prompt be “00011” and the query be “how many zeros are in the prompt?”. The answer should be “3”. If we simply pass “3” as the input to the LLM as the compressed prompt with the query “how many zeros are in the prompt?,” the answer will be “0”, which is incorrect for the given prompt.
>
>     Indeed, we expect that a “good” compressor is able to extract as much of the answer as possible from the prompt and compress that information, but understanding this is orthogonal to our work; we instead find the optimal theoretical performance that can be achieved by *any* compressor (which may or may not use this “find answer, compress” strategy).
> * Q3 (**Clarification on query-aware**):
>     * We wish to clarify a potential misunderstanding: “we assume that...LLM inference call” does not imply the query-aware setting — this is the case for both query-aware and query-agnostic compression (Fig. 2 in paper). The difference in the query-aware case is that the compressor also has access to the query.
>     * The query-aware and query-agnostic settings are both independently important to study theoretically. Our work provides an information-theoretic framework to study hard prompt compression, an area of active research, with several schemes already proposed. These schemes are either query-agnostic (LLMLingua) or query-aware (LLMLingua Query); we cover both types in our framework.
>     * We only develop the theory for the query-agnostic setting in the main paper since the query-aware setting is analogous.
> * Q4 (**Notation in Prop. 1**): For a finite set $A = \\{ a, b, c \\}$, we denote $\\mathbb{R}^{A}_{+}$ as the set of all vectors $x$ with components indexed by elements of $A$, i.e., $x = (x_a, x_b, x_c)$, where $x_a, x_b, x_c$ are nonnegative real numbers. (An overview of our notation is provided in App. A, but we will explain these instances of non-standard notation in the main text.)
> * Q5 (**Constants in Prop. 1**): The constants $\\boldsymbol{D}_{x}$ and $\\boldsymbol{R}_x$ are vectors with nonnegative real components, indexed by elements of $\\mathcal{M}_x$, i.e, ${\\boldsymbol{D}_x}_m$ and ${\\boldsymbol{R}_x}_m$ are the components of $\\boldsymbol{D}_x$ and $\\boldsymbol{R}_x$.
> * Q6 (**Proposed method does not achieve high rate and low distortion?**):
> Our proposed method does match the “No Compression” result when the rate is 1, and can likely achieve low distortion for rates less than 1, but we are not able to exactly characterize when that transition happens for this dataset. LLMLingua-2 Dynamic does not compress with higher rates on our binary dataset, and this is because the $[0, 1]$ confidence scores for each token which should be kept (according to the training data) are very high (between 0.98 and 0.99), and the scores for the tokens which should not be kept are very low. In fact, we could not find a threshold greater than 0 which gives a high rate. For our new natural language experiments, the curves for our LLMLingua-2 Dynamic method fully cover the range of low to high rates.
>
>    We will include updated figures containing at least the RD point for rate 1 in the final revision of our paper, in addition to the discussion above explaining this behavior.
> * Q7 (**Comparisons to other baselines**): Please note that [4] (which we cite as [10] in our paper) is a *soft* prompt compression scheme and is not compatible with our framework, which focuses on *hard prompts*. While methods that use soft prompts are a valid approach to the prompt compression problem, they are not compatible with black-box LLMs, which is a crucial component of our framework.
>
>     Thank you for bringing [5] to our notice; we will cite them in the final version of our paper. Unfortunately, their code is not publicly available; we are unable to integrate their method into our experiments.
> * Q8 (**Additional experiments**): Thank you for this suggestion; we agree that extending our experiments to (ideally large-scale) natural language datasets is important for our paper. We provide the results of two new experiments on natural language data, including NarrativeQA, in our global response.
> ___
> **References**:
>
> [1] J. Ziv and A. Lempel, "Compression of individual sequences via variable-rate coding," in IEEE Trans. Info. Th., 1978
>
> [2] C. T. Li and A. E. Gamal, "Strong Functional Representation Lemma and Applications to Coding Theorems," in IEEE Trans. Info. Th., 2018
>
> [3] C. T. Li and V. Anantharam, "A Unified Framework for One-Shot Achievability via the Poisson Matching Lemma," in IEEE Trans. Info. Th., 2021
>
> [4] D. Wingate, M. Shoeybi, and T. Sorensen. "Prompt compression and contrastive conditioning for controllability and toxicity reduction in language models." arXiv:2210.03162 (2022)
>
> [5] W. Fei, et al. "Extending context window of large language models via semantic compression." arXiv:2312.09571 (2023)

---

> > ### Comment · Reviewer_cHot · 2024-08-12
> >
> > Thank you for the detailed response. Given the modifications the authors promise, I have raised my rating to accept.

---

> > > ### Author Response · Authors · 2024-08-13
> > >
> > > We appreciate the reviewer's timely response and sincerely thank the reviewer for raising their score.

---

### Author Rebuttal · Authors · 2024-08-07

We thank all the reviewers for acknowledging our contributions and providing constructive feedback. Our ***key contributions*** lie in providing a principled framework for prompt compression (as acknowledged by Reviewers cHot and sxbS), showing a large gap between optimality and current schemes (as acknowledged by Reviewers Jydp and sxbS). We are able to do so by formulating the optimal curve as a linear program that can be solved using an efficient, geometric solution via its dual in Algorithm 1. This is nontrivial as the linear program itself has too large a dimension to solve directly. We also propose compression schemes that outperform current schemes on synthetic datasets.

We highlight some of the ***additional experiments*** we ran to address the reviewers’ concerns, which include results on (1) a small-scale dataset with natural language prompts to compare with the optimal compressors on natural language, and (2) a larger scale dataset, NarrativeQA, as suggested by reviewer cHot.

1. **NLP dataset**: We prompt GPT4 to curate a small-scale natural language dataset consisting of short prompts (no more than 15 tokens in length), queries, and their respective answers (some examples from the dataset are shown in Table 1 in our attached PDF). We will include details on the dataset in the final revision of the paper.  We are also able to compute the optimal curve for this small NLP dataset, using the “pruning” approximation described in Appendix E of the paper. Please see Figure 1 in our attached PDF for the results of this experiment. Our key observations are (1) the gap between the optimal prompt compressors and current methods is quite large, (2) Our proposed methods achieve the lowest distortion for rates below 0.5, and (3) with prompt compression, it is possible to do better than standard prompting as shown by the gap between the optimal curves and the “No Compression” result. All of these observations are in line with our experiments on binary sequences.

2. **Extension to a larger scale NLP dataset** (suggested by Reviewers cHot, Jydp, and sxbS): As per the suggestion of Review cHot, we use the summaries provided in the NarrativeQA [1] datasets as the prompts (up to several hundred tokens), and the queries in the dataset accordingly. The result of this experiment is shown in Figure 2 in our attached PDF.
    * Comparison of constructive algorithms: Similar to the result of the small natural language dataset, we observe that our proposed methods again perform better than all other methods for rates less than 0.5.
    * As discussed in item 3 below, it is not feasible to compute the curves for the optimal prompt compressors on this (or any other) large scale dataset. As such, we view this experiment to be complementary to the experiment discussed in item 1 above.

3. **Fundamental characteristics of the optimal curve computation**:  A key observation is that Algorithm 1 can be used to compute the optimal curve even for practical datasets, since it only has a linear complexity in the length of the prompt. The difficulty when assuming black-box models is that we need to make an inference call with each possible compressed prompt to compute its distortion, resulting in an exponential complexity in the length of the prompt. This cannot be avoided without further understanding the structure of language models or making some assumptions on their statistical properties. We leave such considerations, including methods to approximate the optimal curve for larger datasets, for future work.

___
**References**:

[1] Kočiský, Tomáš, et al. "The NarrativeQA Reading Comprehension Challenge." Transactions of the Association for Computational Linguistics 6 (2018): 317-328.

---

> ### Author Response · Authors · 2024-08-14
> **Summary of review and discussion**
>
> Once again, we thank the reviewers for their feedback in the official review and their follow-up during the discussion period. We are delighted that both reviewer cHot and reviewer sxbS raised their scores from 5 to 7 in response to our rebuttal, and are appreciative of reviewer Jydp's initial score of 7, recognizing our contributions as listed in our global response above. The following list summarizes the additions that we will make to the final vision of our manuscript. These additions are based on the comments and questions of the reviewers during the official review, and are already made available in our rebuttal.
>
> 1. We will include our **new experiments** on the small natural language dataset and the large-scale NarrativeQA dataset. All three reviewers requested more experiments on either natural language data, a large-scale dataset, or both. We thank reviewer cHot in particular for the suggestion on using the NarrativeQA dataset.
>
> 2. We will include a table for the timings of each prompt compression method on the small natural language dataset and the NarrativeQA dataset, as a benchmark of **efficiency** of each method. We thank reviewer Jydp for this suggestion.
>
> 3. We will include a discussion on **semantic distortion** and our choice for the distortion functions in our experiments. We thank reviewer sxbS for this suggestion.

---

### Decision · Program_Chairs · 2024-09-25

**Decision:**

Accept (poster)

**Comment:**

The paper provides an interesting rate-distortion framework for fundamental limits of prompt compression for large language models. The reviewers all agree the contribution and novelty. Please make sure to include the new experiments and the table in the camera ready version.